# Efficient and Robust Behavior Policy Search for Online Off-policy Evaluation through Transition Gradients

## Abstract

In reinforcement learning policy evaluation, classic on-policy methods often suffer from high variance when estimating policy performance. To mitigate this issue, *behavior policy search* has been proposed to learn data-collecting policies tailored to reduce online evaluation variance. However, these approaches do not account for uncertainties in the transition functions. In practice, simulator transitions often differ from real world due to modeling errors or approximation limitations. As a result, behavior policies trained in simulation may still yield high variance when deployed in real environments, leading to costly reliance on real-world evaluation samples. In this work, we propose a double-loop gradient-based algorithm for learning behavior policies that are both efficient and robust to transition uncertainty. Theoretically, we derive novel transition-variance gradient expressions and establish global convergence guarantees for the algorithm. Numerically, we demonstrate that our method is less sensitive to transition perturbations than existing approaches, providing supportive evidence for its practical utility.

## 1 Introduction

Reinforcement learning (RL) has achieved remarkable success in recent years across domains such as robotics, healthcare, recommendation systems, and natural language processing (Mnih et al., 2015; Silver et al., 2017; Jumper et al., 2021). A central component in these advances is policy evaluation, the task of estimating the performance of a policy. The most direct approach is the on-policy Monte Carlo method, which collects trajectories from the target policy and estimates its value by averaging the observed returns. Although conceptually simple and widely used, this method often suffers from high evaluation variance, limiting the reliability of the resulting estimates.

To improve efficiency, a growing body of work has investigated learning a separate data-collecting behavior policy to reduce evaluation variance via off-policy evaluation (Hanna et al., 2017; Zhong et al., 2022; Liu and Zhang, 2024; Liu et al., 2025a). This line of research, known as behavior policy search (BPS), optimizes a variance-reducing behavior policy so that the collected trajectories yield more informative evaluations. With an appropriately chosen behavior policy, BPS has been shown to achieve lower variance than naive on-policy evaluation. By contrast, in the standard formulation of off-policy evaluation (OPE), the data are assumed to be pre-logged by a fixed behavior policy and the focus is on designing improved estimators. In comparison, BPS explicitly optimizes the behavior policy itself to reduce variance and is broadly applicable across different off-policy evaluation estimators.

Despite this progress, existing BPS methods generally optimize behavior policies under prespecified transition functions, without accounting for underlying uncertainties. In practice, the true transition functions often deviate from the assumed model due to approximation errors, adversarial perturbations, or partial observability. Such discrepancies can compromise evaluation reliability: a behavior policy optimized under simulator transition may still yield high variance when deployed in the real environment. Consequently, prior methods may continue to require a large number of costly real-world samples to achieve accurate evaluation.

To address these two challenges—variance reduction and transition mismatch—we propose an **efficient and robust policy evaluation** framework that explicitly accounts for transition uncertainty.

Our method formulates BPS as a minimax optimization problem, where an adversarial transition model seeks to maximize the evaluation variance while the behavior policy is optimized to minimize it. Our contributions are summarized as follows: **(1)** We introduce a novel adversarial framework for robust behavior policy search in policy evaluation (Section 3.3); **(2)** We derive analytical transition-gradient expressions for both on-transition and off-transition settings, and provide convergence guarantees for the inner-loop adversarial optimization (Section 4); **(3)** We propose a double-loop robust gradient algorithm and provide global convergence guarantee for variance-minimizing behavior policy search (Section 5); **(4)** We numerically demonstrate that our method is less sensitive to transition perturbations than existing approaches (Section 6), verifying the utility of our theoretical results.

## 2 RELATED WORK

**Behavior Policy Search**

Reducing the variance of policy evaluation in reinforcement learning has been widely studied, with a growing line of work known as behavior policy search (BPS), which optimizes the data-collecting policy to directly reduce variance. Hanna et al. (2017) formulate the task of searching for a variance-reducing behavior policy as an optimization problem. They use stochastic gradient descent to update a parameterized behavior policy, demonstrating that properly chosen behavior policies can achieve lower variance than the on-policy Monte Carlo method. Zhong et al. (2022) further advance this idea by proposing adaptive behavior policies that emphasize under-sampled regions of the state space, thereby improving evaluation efficiency. However, both approaches do not consider the robustness of their learned behavior policy against transition uncertainties. As a result, even if their methods achieve low evaluation variance in the simulator, they might still yield high variance under real-world transition functions. By contrast, our method explicitly accounts for transition uncertainty, ensuring that the learned behavior policy remains effective even under perturbed dynamics.

Liu and Zhang (2024) also study the variance reducing problem, deriving a closed-form, offline-learnable behavior policy with theoretical guarantees. However, their formulation relies on pre-logged data with prespecified and fixed transition probabilities, and cannot adapt when these probabilities shift at deployment, leaving it vulnerable to modeling errors and sim-to-real mismatches. Our method fills this gap by integrating adversarial transition modeling into behavior policy search, combining efficiency with robustness to transition shifts.

**Robust Policy Evaluation**

Recent work has begun addressing robustness in reinforcement learning policy evaluation. For example, Katdare et al. (2023) and Voloshin et al. (2021) propose techniques to improve robustness under simulator mismatch via estimator modification or robust model learning. However, these methods either rely on access to real-world data or focus on minimizing worst-case prediction errors, and do not directly address the high-variance issue central to policy evaluation. In contrast, our work proactively reduces variance by designing behavior policies that are robust to adversarial transition shifts, without requiring target-environment data.

While robust MDP (RMDP) frameworks (Iyengar, 2005; Nilim and El Ghaoui, 2005) also consider robustness under transition uncertainty, they are typically designed for reward maximization and rely on linear programming techniques. These approaches (e.g., Wang et al. (2023a; 2024)) do not apply to our setting, where the goal is to minimize the variance of policy value estimators, a fundamentally non-linear objective. We fill this gap by proposing a novel adversarial transition variance gradient method that explicitly targets variance reduction under transition uncertainty.

## 3 BACKGROUND

### 3.1 MARKOV DECISION PROCESS

We study a finite horizon Markov Decision Process (MDP, Puterman (2014)), which contains a finite action space $\mathcal{A}$, a finite state space $\mathcal{S}$, a transition probability function $p : \mathcal{S} \times \mathcal{A} \to \Delta(\mathcal{S})$, a reward function $r : \mathcal{S} \times \mathcal{A} \to [0, 1]$, an initial state distribution $p_0 : \mathcal{S} \to [0, 1]$, and a fixed horizon length $T$. To simplify notations, we consider the undiscounted setting without loss of generality. Our method naturally applies to the discounted setting as long as the horizon is fixed and finite (Puterman, 2014).

We define $\Delta(\mathcal{X})$ for a finite set $\mathcal{X}$ as the probability simplex over $\mathcal{X}$, i.e., $\Delta(\mathcal{X}) \doteq \{p : \mathcal{X} \to [0,1] \mid \sum_{x \in \mathcal{X}} p(x) = 1\}$. Then, the policy $\pi : \mathcal{S} \to \Delta(\mathcal{A})$ is the function mapping states to probability distribution over the action space $\mathcal{A}$. We consider the parameterized policies $\pi_\theta$, where the parameters $\theta \in \Theta$ is a vector with $\Theta \subseteq \mathbb{R}^n$ for some constant $n$. Likewise, we parameterize the transition function $p_\omega : \mathcal{S} \times \mathcal{A} \to \Delta(\mathcal{S})$ by a parameter $\omega \in \Omega$, where $\Omega \subseteq \mathbb{R}^m$ and is compact. Unless stated otherwise, all norms used in this work are Euclidean (i.e., $\|x\| = \|x\|_2$).

The MDP process begins at time step 0, where an initial state $S_0$ is sampled from $p_0$. At each time step $t \in [T-1]$, an action $A_t$ is sampled based on $\pi(\cdot \mid S_t)$. Then, a finite reward $R_{t+1} \doteq r(S_t, A_t)$ is given by the environment and a successor state $S_{t+1}$ is obtained based on $p(\cdot \mid S_t, A_t)$. After $T$ steps, the agent's interaction with the environment terminates. If the agent reaches any terminal state before time step $T$, it stays there and receives zero reward.

We use $h \doteq \{S_0, A_0, R_1, S_1, A_1, ... S_{T-1}, A_{T-1}, R_T\}$ to denote the trajectory of this MDP. We then define the *return* of $h$ as $g(h) \doteq \sum_{t=0}^{T-1} R_{t+1}$. For any policy, we have a distribution over the trajectory as $\Pr(H = h|\pi)$, where $H$ is a random variable used to denote the trajectory. Lastly, we define the *value* of a policy as $v(\pi) \doteq \mathbb{E}_{H \sim \pi}[g(H)]$.

## 3.2 VARIANCE REDUCTION IN POLICY EVALUATION

We consider the task of reinforcement learning policy evaluation, where the goal is to estimate the value of an interested policy $\pi_e$, called the *target policy*. The traditional *on-policy* Monte Carlo (MC) method estimates $v(\pi_e)$ by repeatedly executing the target policy $\pi_e$ online and averaging the observed returns. That is, $\text{MC}(\pi_e, H) \doteq \frac{1}{n} \sum_{i=0}^{n-1} g(H_i)$ for all $H_i \sim \pi_e$. However, in practice, this straightforward method can induce high evaluation variance, leading to less reliable results (Liu and Zhang, 2024; Liu et al., 2025b;a; Chen et al., 2025).

To mitigate this challenge, recent work has proposed to use *off-policy* evaluation method to reduce variance, where we execute a different policy $\pi_\theta$ (the *behavior policy*), to collect data. For wide applicability, we consider a general off-policy estimator $\text{OPE}(\pi_e, \pi_\theta, H)$, which estimates the value of $\pi_e$ using trajectories $H$ from $\pi_\theta$. A standard example is importance sampling, $\text{IS}(\pi_e, \pi_\theta, H) \doteq g(H) \prod_{t=0}^{T-1} \frac{\pi_e(A_t|S_t)}{\pi_\theta(A_t|S_t)}$. Prior work has shown that with a properly designed behavior policy $\pi_\theta$, one can achieve lower evaluation variance with an off-policy estimator than the traditional on-policy MC method (Hanna et al., 2017; Zhong et al., 2022). This is known as the *Behavior Policy Search* (BPS) problem, where we aim to solve $\min_{\theta \in \Theta} \mathbb{V}_{H \sim \pi_\theta}[\text{OPE}(\pi_e, \pi_\theta, H)]$.

## 3.3 ROBUST BEHAVIOR POLICY SEARCH

Standard behavior policy search methods typically assume fixed transition probabilities, but in practice there are often discrepancies between simulators and real environments. After learning a variance-reducing behavior policy in simulation, practitioners typically deploy it to evaluate the target policy in the real system. However, because *robustness to dynamics shifts* is not considered during the behavior policy search phase, the resulting policy may still induce high variance when collecting real-world data. Consequently, achieving reliable evaluation often requires a large amount of costly real-world samples, motivating a robustness-aware formulation.

To address this, we formulate the robust behavior policy search problem as a minimax optimization:

$$\min_{\theta \in \Theta} \max_{\omega \in \Omega} \mathbb{V}_{H \sim p_\omega, \pi_\theta}[\text{OPE}(\pi_e, \pi_\theta, H)], \tag{1}$$

where the inner maximization identifies worst-case transition perturbations, and the outer minimization seeks a behavior policy that mitigates this adversarial effect. Such min–max formulations are standard in the robust RL literature for modeling adversarial dynamics (Katdare et al., 2023; Voloshin et al., 2021). To further analyze the min-max problem in (1), we can write it as the following equivalent problem

$$\min_{\theta \in \Theta} \{\Phi(\theta) \doteq \max_{\omega \in \Omega} \mathbb{V}_{H \sim p_\omega, \pi_\theta}[\text{OPE}(\pi_e, \pi_\theta, H)]\}, \tag{2}$$

which minimizes the worst-case evaluation variance (Jin et al., 2020). Notably, the function $\Phi$ is not differentiable, and is neither convex nor concave. Thus, we are unable to solve the problem through direct gradient descent on the function $\Phi$, which motivates our double-loop approach (Algorithm 2) in Section 5 with global convergence guarantee.

## 4 SOLVING THE INNER LOOP

In this section, we study the inner loop of the optimization problem (1), which identifies adversarial dynamics that maximize evaluation variance. We derive analytical gradient expressions of the variance with respect to the transition probability, considering both *on-transition* and *off-transition* cases, in analogy to on-policy and off-policy settings. In the on-transition case, the simulator transition can be modified, so trajectories are sampled directly from the evolving $p_\omega$ at each iteration. In the off-transition case, the simulator transition is fixed at $p_{\omega_0}$, and trajectories are collected under this fixed transition probability while reweighted toward the target $p_\omega$. We introduce Algorithm 1, which adapts $p_\omega$ to maximize evaluation variance, serving as the adversarial player against the robust behavior policy $\pi_\theta$ (Section 5). We provide theoretical convergence guarantees for this algorithm. To the best of our knowledge, this is the first work to develop adversarial transition-gradient methods for variance objectives in reinforcement learning.

### 4.1 ON-TRANSITION GRADIENT OF THE VARIANCE

We begin with the *on-transition* case, analogous to the on-policy setting, where the simulator transition can be directly modified to follow the target transition $p_\omega$ at each iteration. Given a fixed behavior policy $\pi_\theta$, we look for the variance-maximizing adversarial transition $p_\omega$. Formally, we need to solve

$$\max_{\omega \in \Omega} \mathbb{V}_{H \sim p_\omega, \pi_\theta} \left[ \text{OPE}(\pi_e, \pi_\theta, H) \right].$$

In the following theorem, we give a gradient expression of the evaluation variance. Importantly, this analytical form is general and applies to any off-policy evaluation estimator, forming the foundation of our transition-gradient method.

**Theorem 1** (Transition Gradient of the Variance). *For a fixed behavior policy $\pi_\theta$,*

$$\frac{\partial}{\partial \omega} \mathbb{V}_{H \sim p_\omega, \pi_\theta}[\text{OPE}(\pi_e, \pi_\theta, H)]$$

$$= \mathbb{E}_{H \sim p_\omega, \pi_\theta} \left[ \text{OPE}(\pi_e, \pi_\theta, H)^2 \sum_{t=0}^{T-1} \frac{\partial}{\partial \omega} \log(p_\omega(S_{t+1}|S_t, A_t)) \right]$$

$$- 2\mathbb{E}_{H \sim p_\omega, \pi_\theta}[\text{OPE}(\pi_e, \pi_\theta, H)] \mathbb{E}_{H \sim p_\omega, \pi_\theta} \left[ \text{OPE}(\pi_e, \pi_\theta, H) \sum_{t=0}^{T-1} \frac{\partial}{\partial \omega} \log(p_\omega(S_{t+1}|S_t, A_t)) \right].$$

Its proof is in Appendix A.2. This gradient expression is in expectation forms and can be estimated unbiasedly from sampled trajectories without additional structural assumptions on the OPE estimator. Building on Theorem 1, we now present the On-transition Variance Gradient method in Algorithm 1. We instantiate our algorithm with the importance sampling estimator (IS), but our framework is ready to accommodate any off-policy evaluation estimator.

To discuss the convergence property of Algorithm 1, we impose the standard Robbins-Monroe step-size condition $\sum_{i=0}^{\infty} \alpha_i = \infty$ and $\sum_{i=0}^{\infty} \alpha_i^2 < \infty$ (Robbins and Monro, 1951). We assume the importance sampling ratio $\frac{\pi_e(a|s)}{\pi_\theta(a|s)}$ exists and is bounded above for all $s$, $a$, and $\theta$ (Hanna et al., 2024). Besides, we require the transition $p_\omega$ to be twice-differentiable with respect to $\omega$ with uniformly bounded first- and second-order derivatives. These conditions for $p_\omega$ hold, for example, when it is parameterized by a neural network with smooth activations and a softmax output layer, and are commonly adopted in policy gradient literature (e.g., Hanna et al. (2017; 2024)). Then, we have the following lemma for the convergence of Algorithm 1, whose proof is in Appendix A.3.

**Lemma 1** (Transition Gradient Convergence). *For a fixed behavior policy $\pi_\theta$, Algorithm 1 converges. That is, $\mathbb{V}_{H_i \sim p_{\omega_i}, \pi_\theta}[\text{IS}(\pi_e, \pi_\theta, H_i)]$ converges to a finite value and*
$\lim_{i \to \infty} \frac{\partial}{\partial \omega} \mathbb{V}_{H_i \sim p_{\omega_i}, \pi_\theta}[\text{IS}(\pi_e, \pi_\theta, H_i)] = 0.$

In practice, although discrepancies often exist between the transition probability in the deployment environment and the original simulator, the simulator typically remains a reasonable approximation. Thus, to ensure the learned adversarial transition remains realistic rather than overly pessimistic, we also offer an optional Kullback–Leibler (KL) divergence penalty that discourages large deviations between $p_\omega$ and the initial simulator transition $p_{\omega_0}$. Given a behavior policy $\pi_\theta$, we consider the

---

**Algorithm 1:** On-Transition Variance Gradient.

1: **Input:** an initial transition parameter $\omega_0$, a target policy $\pi_e$, a fixed behavior policy $\pi_\theta$, a number of iteration $n$, a batch size $k$, a step-size $\alpha_i$ for each $i$
2: **Output:** a final adversarial transition parameter $\omega_n$
3: **For all** $i \in 0, ..., n-1$ **do**
4:    Sample $k$ trajectories $H \sim \pi_\theta, p_{\omega_i}$
5:    $\omega_{i+1} = \omega_i + \frac{\alpha_i}{k}\Big[ \sum_{j=1}^{k} \left( \text{IS}(\pi_e, \pi_\theta, H^j)^2 \sum_{t=0}^{T-1} \frac{\partial}{\partial\omega} \log\big(p_{\omega_i}^j(S_{t+1}|S_t, A_t)\big)\right) -$
    $2\sum_{j=1}^{\frac{k}{2}} \text{IS}(\pi_e, \pi_\theta, H^j) \sum_{j=\frac{k}{2}+1}^{k} \left( \text{IS}(\pi_e, \pi_\theta, H^j) \sum_{t=0}^{T-1} \frac{\partial}{\partial\omega} \log\big(p_{\omega_i}^j(S_{t+1}|S_t, A_t)\big)\right)\Big]$
6: **End for**
7: **Return:** $\omega_n$

---

following inner-loop optimization problem under KL regularization:

$$\max_{\omega\in\Omega}\mathbb{V}_{H\sim p_\omega,\pi_\theta}\left[\text{OPE}(\pi_e, \pi_\theta, H)\right] - \eta D_{\text{KL}}(\Pr(H|p_\omega)\|\Pr(H|p_{\omega_0})),$$

where $\eta > 0$ is the regularization coefficient and the KL-divergence term is defined as $D_{\text{KL}}(\Pr(H|p_\omega)\|\Pr(H|p_{\omega_0})) \doteq \mathbb{E}_{H\sim p_\omega,\pi_\theta}\left[\log\frac{\Pr(H|p_\omega)}{\Pr(H|p_{\omega_0})}\right]$. To simplify notations, we define $\ell_{p_\omega} \doteq \sum_{t=0}^{T-1}\log(p_\omega(S_{t+1}|S_t, A_t))$. We provide the gradient expression of this regularized optimization problem in the following theorem.

**Theorem 2** (Transition Gradient of Variance with KL). *For a fixed behavior policy $\pi_\theta$ and a regularization coefficient $\eta > 0$,*

$$\frac{\partial}{\partial\omega}\mathbb{V}_{H\sim p_\omega,\pi_\theta}[\text{OPE}(\pi_e, \pi_\theta, H)] - \eta D_{\text{KL}}(\Pr(H|p_\omega)\|\Pr(H|p_{\omega_0}))$$
$$=\mathbb{E}_{H\sim p_\omega,\pi_\theta}\left[\text{OPE}(\pi_e, \pi_\theta, H)^2\frac{\partial}{\partial\omega}\ell_{p_\omega}\right]$$
$$- 2\mathbb{E}_{H\sim p_\omega,\pi_\theta}[\text{OPE}(\pi_e, \pi_\theta, H)]\mathbb{E}_{H\sim p_\omega,\pi_\theta}\left[\text{OPE}(\pi_e, \pi_\theta, H)\frac{\partial}{\partial\omega}\ell_{p_\omega}\right]$$
$$- \eta\mathbb{E}_{H\sim p_\omega,\pi_\theta}\left[\left(\frac{\partial}{\partial\omega}\ell_{p_\omega}\right)\left(1 + \ell_{p_\omega} - \ell_{p_{\omega_0}}\right)\right].$$

It proof is in Appendix A.4. This regularization balances robustness with realism, ensuring that the learned adversary remains close to plausible dynamics.

### 4.2 OFF-TRANSITION GRADIENT OF THE VARIANCE

In the *off-transition* case, analogous to the off-policy setting, simulator transitions are fixed at $p_{\omega_0}$ and may differ from the target transition $p_\omega$. This situation arises naturally when using black-box simulators that permit data collection but do not allow modifying transition probabilities (e.g., Komorowski et al. (2018)). In this case, we introduce a transition importance sampling ratio to reweight the collected trajectories, mirroring the familiar correction used in off-policy evaluation for policies. For a general off-policy estimator OPE, we overload the notation as

$$\text{OPE}(\pi_e, \pi_\theta, p_\omega, H) \doteq \frac{\prod_{t=0}^{T-1} p_\omega(S_{t+1}|S_t, A_t)}{\prod_{t=0}^{T-1} p_{\omega_0}(S_{t+1}|S_t, A_t)}\text{OPE}(\pi_e, \pi_\theta, H).$$

We omit the input $p_{\omega_0}$ in $\text{OPE}(\pi_e, \pi_\theta, p_\omega, H)$ to simplify notations. Similar to Theorem 1, we first give an analytical gradient expression of the evaluation variance.

**Theorem 3** (Off-Transition Gradient of Variance). *When $p_\omega \neq p_{\omega_0}$, for a fixed behavior policy $\pi_\theta$,*

$$\frac{\partial}{\partial\omega}\mathbb{V}_{H\sim p_{\omega_0},\pi_\theta}[\text{OPE}(\pi_e, \pi_\theta, p_\omega, H)]$$
$$=2\mathbb{E}_{H\sim p_{\omega_0},\pi_\theta}\left[\text{OPE}^2(\pi_e, \pi_\theta, p_\omega, H)\frac{\partial}{\partial\omega}\ell_{p_\omega}\right]$$
$$- 2\mathbb{E}_{H\sim p_{\omega_0},\pi_\theta}[\text{OPE}(\pi_e, \pi_\theta, p_\omega, H)] \cdot \mathbb{E}_{H\sim p_{\omega_0},\pi_\theta}\left[\text{OPE}(\pi_e, \pi_\theta, p_\omega, H)\frac{\partial}{\partial\omega}\ell_{p_\omega}\right].$$

Its proof is in Appendix A.5. In the next theorem, we incorporate a KL-divergence term to penalize large deviations of $p_\omega$ from the simulator's transition $p_{\omega_0}$, with the KL direction chosen so that the expectation aligns with the available sampling distribution $p_{\omega_0}$. This design ensures the realism of the learned adversarial transition.

**Theorem 4** (Off-transition Gradient of Variance with KL). *For a fixed behavior policy $\pi_\theta$ and a regularization coefficient $\eta > 0$,*

$$\frac{\partial}{\partial \omega} \mathbb{V}_{H \sim p_{\omega_0}, \pi_\theta}[\mathrm{OPE}(\pi_e, \pi_\theta, p_\omega, H)] - \eta D_{\mathrm{KL}}(\Pr(H|p_{\omega_0}) \| \Pr(H|p_\omega))$$

$$= 2\mathbb{E}_{H \sim p_{\omega_0}, \pi_\theta} \left[ \mathrm{OPE}^2(\pi_e, \pi_\theta, p_\omega, H) \frac{\partial}{\partial \omega} \ell_{p_\omega} \right]$$

$$\quad - 2\mathbb{E}_{H \sim p_{\omega_0}, \pi_\theta}[\mathrm{OPE}(\pi_e, \pi_\theta, p_\omega, H)] \mathbb{E}_{H \sim p_{\omega_0}, \pi_\theta} \left[ \mathrm{OPE}(\pi_e, \pi_\theta, p_\omega, H) \frac{\partial}{\partial \omega} \ell_{p_\omega} \right]$$

$$\quad - \eta \mathbb{E}_{H \sim p_{\omega_0}, \pi_\theta} \left[ -\frac{\partial}{\partial \omega} \ell_{p_\omega} \right].$$

Its proof is in Appendix A.6. Note that the gradient expression in the off-transition setting (Theorem 4) differs from that in the on-transition case (Theorem 2), reflecting the distinct data sampling mechanisms.

## 5 SOLVING THE OUTER LOOP

In this section, we propose a behavior policy search (BPS) method that is robust to potential discrepancies in the environment. Specifically, we adopt a policy gradient approach to search for a variance-reducing behavior policy under an adversarial transition probability. We first introduce this algorithm, theoretically analyzing its global convergence guarantee. Then, in Section 6, we demonstrate its empirical robustness under perturbed transition probabilities.

### 5.1 DOUBLE-LOOP ROBUST VARIANCE GRADIENT

To begin with, recall that in (1), our goal is to solve the min-max objective

$$\min_{\theta \in \Theta} \max_{\omega \in \Omega} \mathbb{V}_{H \sim p_\omega, \pi_\theta} \left[ \mathrm{OPE}(\pi_e, \pi_\theta, H) \right].$$

In Section 4 and Algorithm 1, we present methods to solve the inner maximization problem by performing gradient ascent on the transition parameter $\omega$. In this section, we focus on performing gradient descent for the variance objective on the policy parameter $\theta$. This is also known as the *behavior policy search* problem in off-policy evaluation (OPE) community (Hanna et al., 2017; 2024), which aims at finding a variance minimizing behavior policy to collect data through gradient based methods. In Lemma 2, we present the gradient expression for variance with respect to the behavior policy adopted from Hanna et al. (2017).

**Lemma 2** (Variance Gradient Expression). *With a fixed transition probability $p_\omega$, $\forall \theta$,*

$$\frac{\partial}{\partial \theta} \mathbb{V}_{H \sim p_\omega, \pi_\theta}[\mathrm{IS}(\pi_e, \pi_\theta, H)] = \mathbb{E}_{H \sim p_\omega, \pi_\theta}[-\mathrm{IS}(\pi_e, \pi_\theta, H)^2 \sum_{t=0}^{T-1} \frac{\partial}{\partial \theta} \log \pi_\theta(A_t|S_t)].$$

Importantly, this lemma shows that we can estimate the gradient with trajectories sampled from the behavior policy $\pi_\theta$. With this analytical expression, we are now ready to present our double loop algorithm, named *Double-Loop Robust Gradient for Variance* (DRVG).

---

**Algorithm 2:** Double-Loop Robust Variance Gradient (DRVG)

---

1: **Input:** a target policy parameter $\theta_e$, a number of iteration $n$, a batch size $k$, a step-size $\alpha$, tolerance sequence $\{\epsilon_i\}$
2: **Output:** a final robust behavior policy parameter $\theta^*$
3: **For all** $i = 0, ..., n-1$ **do**
4:     Find $p_{\omega_i}$ s.t. $\mathbb{V}_{H \sim p_{\omega_i}, \pi_{\theta_i}}[\mathrm{IS}(\pi_e, \pi_{\theta_i}, H)] \geq \max_{p_\omega} \mathbb{V}_{H \sim p_\omega, \pi_{\theta_i}}[\mathrm{IS}(\pi_e, \pi_{\theta_i}, H)] - \epsilon_i$
5:     $\mathcal{G}_i = \frac{\partial}{\partial \theta} \mathbb{V}_{H \sim p_{\omega_i}, \pi_{\theta_i}}[\mathrm{IS}(\pi_e, \pi_{\theta_i}, H)]$
6:     $\theta_{i+1} = \mathrm{Proj}_\Theta[\theta_i - \alpha \mathcal{G}_i]$
7: **End for**
8: **Return:** $\bar{\theta} \doteq \frac{1}{n} \sum_{i=0}^{n-1} \theta_i$.

---

The double loop algorithm DRVG iteratively takes gradient steps on the evaluation variance objective to solve the min-max problem in (1). Specifically, the inner loop of DRVG returns a worst-case

transition probability $p_{\omega_i}$ up to a precision $\epsilon_i$, which can be obtained through Algorithm 1. Such a sequence $\{\epsilon_i\}$ introduces more flexibility to this double-loop algorithm, allowing for quick policy updates without hurting the global convergence property. This choice is also adopted by some prior work in the robust MDP community (Ho et al., 2021; Wang et al., 2023a).

In the outer loop, DRVG takes a *projected gradient step* to minimize evaluation variance within the feasible parameter set $\Theta$. A well-known proximal representation of projected gradient in Bertsekas (1995) is $\theta_{i+1} \in \arg\min_{\theta \in \Theta} \langle \mathcal{G}_i, \theta - \theta_i \rangle + \frac{1}{2\alpha_i} \|\theta - \theta_i\|^2 = \text{Proj}_{\Theta}[\theta_i - \alpha \mathcal{G}_i]$, where $\text{Proj}_{\Theta}$ is the projection operator onto $\Theta$. In other words, it is identical to taking a plain gradient step, and then using the closest feasible point in Euclidean distance within the feasible set. Notably, when the feasible set $\Theta$ is convex, this projected gradient step can be implemented by a convex optimization solver with a quadratic objective (Wang et al., 2023a). Together, this double-loop algorithm yields a robust behavior policy for off-policy evaluation under environment uncertainty.

## 5.2 GLOBAL CONVERGENCE ANALYSIS

In this subsection, we present the global convergence analysis for Algorithm 2. For the widely-studied policy gradient methods in reinforcement learning *policy improvement*, the objective function is the *performance* of a given target policy. In our *policy evaluation* setting, however, in order to minimize the ultimate online samples needed in the real-world evaluation, the objective function is the *performance's variance*,

$$\mathbb{V}_{H \sim p_{\omega}, \pi_{\theta}}[\text{OPE}(\pi_e, \pi_{\theta}, H)] = \mathbb{E}_{H \sim p_{\omega}, \pi_{\theta}}[\text{OPE}(\pi_e, \pi_{\theta}, h)^2] - \mathbb{E}_{H \sim p_{\omega}, \pi_{\theta}}[\text{OPE}(\pi_e, \pi_{\theta}, h)]^2.$$

The non-linear nature of this variance objective introduces additional difficulties, making the min-max optimization problem (1) nonconvex-nonconcave, which is widely known to be challenging (Jin et al., 2020; Nouiehed et al., 2019; Lin et al., 2020). Besides, the objective function $\Phi(\theta)$ in the equivalent expression (2) is generally non-differentiable and nonconvex, making the theoretical analysis to our Algorithm 2 even more challenging. In fact, even without the inner minimization problem, finding the global optima of such nonconvex objectives is already NP-hard in the worst case (Jin et al., 2020).

In *policy improvement* regime without robustness consideration (i.e., a single-loop performance maximization problem), recent work has shown that some algorithms are guaranteed to converge to a globally-optimal policy with a non-convex objective function in *tabular* MDPs (Agarwal et al., 2021; Bhandari and Russo, 2021). When robustness is introduced via a min–max formalization, only recently was the first generic algorithm with global convergence proposed (Wang et al., 2023a). However, since their inner maximization objective (policy performance) reduces to a linear program in each update, the setting is considerably simpler than our variance-based objective.

In Section 6, we demonstrate the empirical performance of our Algorithm 2 under *a neural network policy parameterization*. While in this section, for the theoretical analysis of Algorithm 2, we adopt a linear-softmax parameterization for the behavior policy $\pi_{\theta}$, $\pi_{\theta}(a|s) \doteq \frac{\exp(\theta_a^{\top} \phi(s))}{\sum_{a' \in \mathcal{A}} \exp(\theta_{a'}^{\top} \phi(s))}$, where $\phi: s \to \mathbb{R}^d$ is a state feature function, and $\theta_a \in \mathbb{R}^d$ is the parameter associated with action $a \in \mathcal{A}$. In this section, we assume that the parameters' feasible set $\Theta$ to be closed and convex with a diameter $D$ (i.e., $\forall \theta, \theta' \in \Theta, \|\theta - \theta'\| \leq D$), and assume the linear feature to be bounded (i.e., $\forall s, \|\phi(s)\| \leq B$ for $B \in \mathbb{R}$) . This choice enables generalization across states through shared features, and makes the variance objective convex in $\theta$. This assumption has been widely adopted in recent theoretical work on policy gradient (Agarwal et al., 2021; Yuan et al., 2022; Cayci et al., 2024).

With the smoothness of this linear-softmax parameterization, we first establish Lemma 3, which characterizes the behavior of the objective function $\mathbb{V}$ with respect to the policy parameter $\theta$. This lemma then helps to derive the Lipschitz continuity and convexity of the otherwise non-convex and non-differentiable objective function $\Phi$ in (2).

**Lemma 3.** *Under linear-softmax policy parameterization, the objective function $\mathbb{V}_{H \sim p_{\omega}, \pi_{\theta}}[\text{IS}(\pi_e, \pi_{\theta}, H)]$ is $L_{\Theta}$-Lipschitz, $\ell_{\Theta}$-smooth, and convex in $\theta$ with $L_{\Theta} = \sqrt{2}BC^{2T}T^3$ and $\ell_{\Theta} = B^2 C^{2T} T^3 (5 + 8T)$, where $C$ denotes a uniform upper bound on the individual importance sampling ratio with $\frac{\pi_e(a|s)}{\pi_{\theta}(a|s)} \leq C, \forall(s, a), \forall \theta \in \Theta$.*

Its proof is in Appendix A.7. While the theoretical constants scale with horizon $T$, this dependence is intrinsic to importance sampling–based approaches and has also appeared in prior OPE analyses (Liu

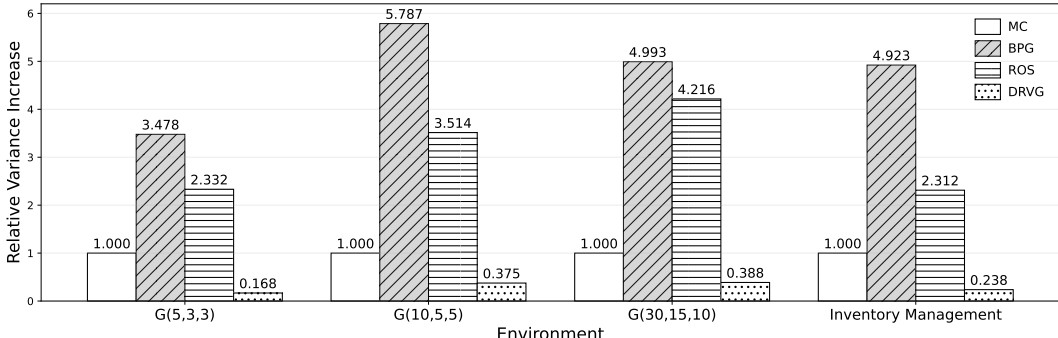

Figure 1: Relative variance increase of each method under its tailored adversarial transition, compared to the variance under the original simulator transition. All values are normalized by the variance increase of the on-policy Monte Carlo (MC) method in the same environment. More details are provided in Appendix B.

et al., 2018; 2020). Our result further shows that global convergence still holds with explicit finite-sample guarantees despite this scaling. With Lemma 3, we can further obtain the desired properties of $\Phi$, which serve as key building blocks for establishing the global convergence of Algorithm 2.

**Lemma 4.** *The function $\Phi(\theta) \doteq \max_{p_\omega} \mathbb{V}_{H \sim p_\omega, \pi_\theta}[\mathrm{IS}(\pi_e, \pi_\theta, H)]$ is $L_\Theta$-Lipschitz and convex in $\theta$.*

Its proof is in Appendix A.8. Equipped with Lemma 3 and Lemma 4, we are now ready to establish the convergence analysis despite the inherent nondifferentiability. The following theorem provides a finite-sample convergence guarantee for our double-loop algorithm.

**Theorem 5** (Double loop global convergence). *With a constant step size $\alpha \doteq \frac{D}{L_\Theta \sqrt{n}}$, we have*

$$\Phi(\bar{\theta}) - \min_{\theta \in \Theta} \Phi(\theta) \leq \frac{D L_\Theta}{\sqrt{n}} + \frac{1}{n} \sum_{i=0}^{n-1} \epsilon_i.$$

Its proof is in Appendix A.9. This result shows that Algorithm 2 converges to an $\epsilon-$optimal solution at a rate of $\mathcal{O}(\frac{1}{\sqrt{n}})$, where $n$ is the number of iterations. This rate matches the optimal rate of projected gradient descent in convex optimization, although our min–max variance objective is more challenging than the performance-based objectives studied in prior work (Agarwal et al., 2021; Bhandari and Russo, 2021; Wang et al., 2023a). The error bound consists of two parts: the first term $\frac{D L_\Theta}{\sqrt{n}}$ reflects the convergence rate of projected gradient descent, while the second term $\frac{1}{n} \sum_{i=0}^{n-1} \epsilon_i$ accounts for the chosen precision in the inner maximization. To our knowledge, this is the first global convergence guarantee for variance-minimizing behavior policy search under adversarial transitions, filling an important gap between classical off-policy evaluation and robust reinforcement learning.

## 6 NUMERICAL RESULTS

In this section, we provide numerical results to validate the utility of our proposed efficient and robust evaluation framework. Our primary goal is to examine two key questions: **(1)** Is our method robust to adversarial transition perturbations? **(2)** Does it give lower evaluation variance under perturbed transitions compared with standard on-policy Monte Carlo?

We evaluate these questions on two environments. Garnet MDPs (Archibald et al., 1995) provide a class of randomly generated abstract MDPs that allow controlled investigation of robustness properties. A Garnet instance $G(|\mathcal{S}|, |\mathcal{A}|, b)$ is parameterized by the number of states $|\mathcal{S}|$, number of actions $|\mathcal{A}|$, and a branching factor $b$ that controls the connectivity of transitions. Owing to this flexibility, Garnets are a standard setting for analyzing robustness in controlled MDP studies (Tarbouriech and Lazaric, 2019; Wang et al., 2023a;b). Inventory management (Porteus, 2002; Ho et al., 2018) is a classical stochastic control problem where a retailer makes ordering decisions under uncertain demand. It provides a natural testbed for evaluating policy performance under transition uncertainties.

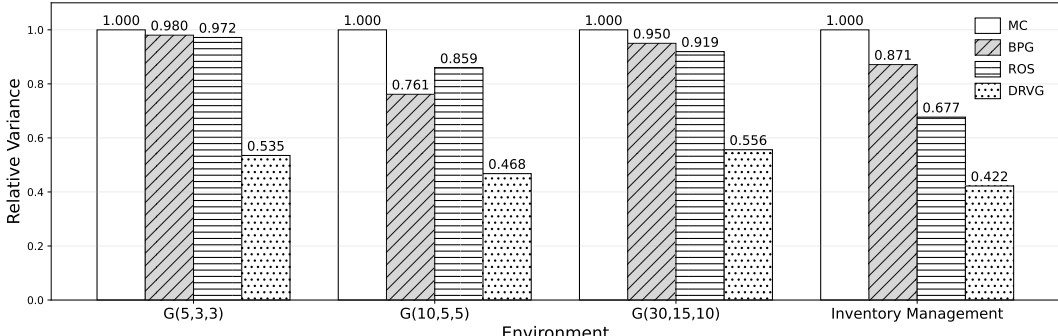

Figure 2: Relative variance of each method under the same perturbed transition. All values are normalized by the variance of the on-policy Monte Carlo (MC) method in the same environment.

To contextualize the results, we compare our approach with several representative methods: the on-policy Monte Carlo estimator (MC), the behavior policy gradient estimator (BPG, Hanna et al. (2017)), and the robust on-policy sampling estimator (ROS, Zhong et al. (2022)). All methods are trained with the same initial transition function to obtain their behavior policies. We parameterize our behavior policy with a neural network and use the final iterate behavior policy from Algorithm 2 to collect evaluation data. Further experimental details are provided in Appendix B.

### 6.1 VARIANCE INCREASE UNDER TAILORED ADVERSARIAL TRANSITIONS

To answer the first question, we examine the robustness of each behavior policy when exposed to its own most adversarial transition. For each method, we run Algorithm 1 to obtain the transition that maximizes its evaluation variance. Then, we use each behavior policy to collect data under its *method-specific adversarial transition*. We report the relative variance increase compared to the original simulator transition, highlighting each method's vulnerability to adversarial perturbations. As shown in Figure 1, our method (DRVG) exhibits the smallest variance increase, illustrating its robustness to adversarial transitions. Notably, although designed for variance reduction, BPG and ROS incur larger variance increases than the on-policy Monte Carlo baseline when the deployment transition is perturbed, underscoring the necessity of our robustness-aware behavior policy search framework.

### 6.2 VARIANCE COMPARISON UNDER SHARED AND PERTURBED TRANSITION

To address the second question, we evaluate all methods under a shared adversarial target transition identified by Algorithm 1 for the on-policy baseline. We compare the variance of all four methods under this *same perturbed transition*. This setup contrasts with Section 6.1, where each method faced its own tailored adversary. As shown in Figure 2, our method (DRVG) indeed achieves lower evaluation variance. This demonstrates that explicitly accounting for transition uncertainty enables more reliable policy evaluation under perturbed environments.

## 7 CONCLUSION

In this work, we present an efficient and robust behavior policy search framework that tackles two central challenges in real-world policy evaluation: variance reduction and transition mismatch. Our method learns variance-reducing behavior policies while explicitly accounting for transition uncertainty through a minimax formulation over adversarial dynamics. Theoretically, we derive novel transition-variance gradient expressions, establish convergence guarantees for the adversarial inner loop, and prove global convergence of our proposed double-loop algorithm. Numerically, our method demonstrates increased robustness under transition perturbations. Taken together, these results unify variance reduction with robustness to transition shifts, offering a promising step toward reliable policy evaluation under uncertainty.

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

# A  PROOF

## A.1  DEFINITIONS

In this section, we show the standard optimization definitions used in our work. Consider an optimization problem

$$\min_{x \in \mathcal{X}} f(x)$$

where $\mathcal{X} \subseteq \mathbb{R}^d$ is nonempty and closed, and $f : \mathbb{R}^d \to \mathbb{R}$. We have the following definitions for Lipschitz continuity and smoothness.

**Definition 1** (Lipschitz Continuity)**.** The function $f : \mathcal{X} \to \mathbb{R}$ is $L-$Lipschitz if $\forall x_1, x_2 \in \mathcal{X}$,

$$\|f(x_1) - f(x_2)\| \leq L\|x_1 - x_2\|.$$

**Definition 2** (Smoothness)**.** The function $f : \mathcal{X} \to \mathbb{R}$ is $\ell-$smooth if $\forall x_1, x_2 \in \mathcal{X}$,

$$\|\nabla f(x_1) - \nabla f(x_2)\| \leq \ell\|x_1 - x_2\|.$$

With these definitions in hand, we are now ready to present the proofs.

## A.2  PROOF OF THEOREM 1

**Theorem 1** (Transition Gradient of the Variance)**.** *For a fixed behavior policy $\pi_\theta$,*

$$\frac{\partial}{\partial \omega} \mathbb{V}_{H \sim p_\omega, \pi_\theta}[\text{OPE}(\pi_e, \pi_\theta, H)]$$

$$= \mathbb{E}_{H \sim p_\omega, \pi_\theta}\left[\text{OPE}(\pi_e, \pi_\theta, H)^2 \sum_{t=0}^{T-1} \frac{\partial}{\partial \omega} \log(p_\omega(S_{t+1}|S_t, A_t))\right]$$

$$- 2\mathbb{E}_{H \sim p_\omega, \pi_\theta}[\text{OPE}(\pi_e, \pi_\theta, H)]\mathbb{E}_{H \sim p_\omega, \pi_\theta}\left[\text{OPE}(\pi_e, \pi_\theta, H) \sum_{t=0}^{T-1} \frac{\partial}{\partial \omega} \log(p_\omega(S_{t+1}|S_t, A_t))\right].$$

*Proof.* To prove Theorem 1, we aim at decomposing the term $\Pr(H = h \mid p_\omega)$ into two parts: one that depends on $p_\omega$ and one that does not. Let

$$m_{p_\omega}(h) \doteq \prod_{t=0}^{T-1} p_\omega(S_{t+1}|S_t, A_t) \tag{3}$$

and

$$p(h) \doteq \frac{\Pr(H = h \mid p_\omega)}{m_{p_\omega}(h)}, \tag{4}$$

then we have

$$\Pr(H = h \mid p_\omega) = p(h)m_{p_\omega}(h). \tag{5}$$

Next, we manipulate the term $\frac{\partial}{\partial\theta}m_{p_\omega}(h)$.

$$
\begin{aligned}
\frac{\partial}{\partial\theta}m_{p_\omega}(h) =& \frac{\partial}{\partial\omega}\prod_{t=0}^{T-1}p_\omega(S_{t+1}|S_t, A_t)\\
=& \sum_{t=0}^{T-1}\left(\prod_{i\neq t}p_\omega(S_{i+1}|S_i, A_i)\frac{\partial p_\omega(S_{t+1}|S_t, A_t)}{\partial\omega}\right)\\
=& \sum_{t=0}^{T-1}\left(\frac{\prod_{i=0}^{T-1}p_\omega(S_{i+1}|S_i, A_i)}{p_\omega(S_{t+1}|S_t, A_t)}\cdot\frac{\partial p_\omega(S_{t+1}|S_t, A_t)}{\partial\omega}\right)\\
=& \prod_{i=0}^{T-1}p_\omega(S_{i+1}|S_i, A_i)\cdot\sum_{t=0}^{T-1}\left(\frac{1}{p_\omega(S_{t+1}|S_t, A_t)}\frac{\partial p_\omega(S_{t+1}|S_t, A_t)}{\partial\omega}\right)\\
\overset{(a)}{=}& \prod_{i=0}^{T-1}p_\omega(S_{i+1}|S_i, A_i)\cdot\sum_{t=0}^{T-1}\left(\frac{1}{p_\omega(S_{t+1}|S_t, A_t)}p_\omega(S_{t+1}|S_t, A_t)\frac{\partial\log p_\omega(S_{t+1}|S_t, A_t)}{\partial\omega}\right)\\
=& \prod_{i=0}^{T-1}p_\omega(S_{i+1}|S_i, A_i)\sum_{t=0}^{T-1}\left(\frac{\partial}{\partial\omega}\log p_\omega(S_{t+1}|S_t, A_t)\right)\\
=& m_{p_\omega}(h)\sum_{t=0}^{T-1}\left(\frac{\partial}{\partial\omega}\log p_\omega(S_{t+1}|S_t, A_t)\right)\\
=& m_{p_\omega}(h)\sum_{t=0}^{T-1}\frac{\partial}{\partial\omega}\log(p_\omega(S_{t+1}|S_t, A_t)) \qquad\qquad (6)
\end{aligned}
$$

Here, (a) follows from the fact that

$$
\begin{aligned}
\frac{\partial}{\partial x}\log f(x) =& \frac{1}{f(x)}\frac{\partial f(x)}{\partial x}\\
\implies \frac{\partial f(x)}{\partial x} =& f(x)\cdot\frac{\partial\log f(x)}{\partial x}.
\end{aligned}
$$

Then, we decompose the variance objective

$$
\begin{aligned}
&\frac{\partial}{\partial\omega}\mathbb{V}_{H\sim p_\omega, \pi_\theta}[\mathrm{OPE}(\pi_e, \pi_\theta, H)]\\
=&\frac{\partial}{\partial\omega}\left(\mathbb{E}_{H\sim p_\omega, \pi_\theta}[\mathrm{OPE}(\pi_e, \pi_\theta, H)^2] - \mathbb{E}_{H\sim p_\omega, \pi_\theta}[\mathrm{OPE}(\pi_e, \pi_\theta, h)]^2\right)\\
=&\frac{\partial}{\partial\omega}\sum_h \Pr(H = h|p_\omega)\mathrm{OPE}(\pi_e, \pi_\theta, h)^2\\
&- 2\mathbb{E}_{H\sim p_\omega, \pi_\theta}[\mathrm{OPE}(\pi_e, \pi_\theta, H)]\frac{\partial}{\partial\omega}\sum_h \Pr(H = h|p_\omega)\mathrm{OPE}(\pi_e, \pi_\theta, h)\\
=&\sum_h p(h)\mathrm{OPE}(\pi_e, \pi_\theta, h)^2\frac{\partial}{\partial\omega}m_{p_\omega}(h)\\
&- 2\mathbb{E}_{H\sim p_\omega, \pi_\theta}[\mathrm{OPE}(\pi_e, \pi_\theta, H)]\sum_h p(h)\mathrm{OPE}(\pi_e, \pi_\theta, h)\frac{\partial}{\partial\omega}m_{p_\omega}(h) \qquad\text{(By (5))}
\end{aligned}
$$

$$= \sum_h p(h) \text{OPE}(\pi_e, \pi_\theta, h)^2 m_{p_\omega}(h) \sum_{t=0}^{T-1} \frac{\partial}{\partial \omega} \log(p_\omega(S_{t+1}|S_t, A_t))$$

$$- 2\mathbb{E}_{H \sim p_\omega, \pi_\theta}[\text{OPE}(\pi_e, \pi_\theta, H)] \sum_h p(h)\text{OPE}(\pi_e, \pi_\theta, h)m_{p_\omega}(h) \sum_{t=0}^{T-1} \frac{\partial}{\partial \omega} \log(p_\omega(S_{t+1}|S_t, A_t))$$

(By (6))

$$= \mathbb{E}_{H \sim p_\omega, \pi_\theta}[\text{OPE}^2(\pi_\theta, H) \sum_{t=0}^{T-1} \frac{\partial}{\partial \omega} \log(p_\omega(S_{t+1}|S_t, A_t))]$$

$$- 2\mathbb{E}_{H \sim p_\omega, \pi_\theta}[\text{OPE}(\pi_e, \pi_\theta, H)]\mathbb{E}_{H \sim p_\omega, \pi_\theta}\left[\text{OPE}(\pi_e, \pi_\theta, H) \sum_{t=0}^{T-1} \frac{\partial}{\partial \omega} \log(p_\omega(S_{t+1}|S_t, A_t))\right].$$

$\square$

### A.3 Proof of Lemma 1

**Lemma 1** (Transition Gradient Convergence). *For a fixed behavior policy $\pi_\theta$, Algorithm 1 converges. That is, $\mathbb{V}_{H_i \sim p_{\omega_i}, \pi_\theta}[\text{IS}(\pi_e, \pi_\theta, H_i)]$ converges to a finite value and $\lim_{i \to \infty} \frac{\partial}{\partial \omega} \mathbb{V}_{H_i \sim p_{\omega_i}, \pi_\theta}[\text{IS}(\pi_e, \pi_\theta, H_i)] = 0.$*

*Proof.* The proof leverages Proposition 3 in Bertsekas and Tsitsiklis (2000), for which we have to show that Algorithm 1 satisfies the following conditions:

1. $\mathbb{V}[\text{IS}(\pi_\theta, p_{\omega_i}, H_i)]$ is continuously differentiable w.r.t. $\omega$.

2. The gradient of the variance objectives, $\frac{\partial}{\partial \omega} \mathbb{V}[\text{IS}(\pi_\theta, p_{\omega_i}, H_i)]$, is Lipschitz continuous w.r.t. $\omega$.

3. The variance of the gradient estimate used by Algorithm 1 is bounded.

The other conditions of Proposition 3 in Bertsekas and Tsitsiklis (2000) are satisfied because of the unbiasedness of the gradient estimates in Algorithm 1. Additionally, since the gradient objective, as a variance, is bounded below by zero, we can avoid the case of converging to $-\infty$ according to Proposition 3 (Bertsekas and Tsitsiklis, 2000).

By assumptions, we have $p_\omega$ is twice-differentiable, and quotient $\frac{w_{\pi_e}}{w_{\pi_\theta}}$ and the estimator $\text{IS}(\pi_\theta, p_\omega, H)$ always exist. Therefore, by the gradient expression in Lemma 1, we conclude that $\frac{\partial}{\partial \omega} V_{H \sim p_\omega, \pi_\theta}[\text{IS}(\pi_\theta, p_\omega, H)]$ is continuously differentiable, verifying condition 1.

Next, we show the Lipschitz continuity of $\frac{\partial}{\partial \omega} V_{H \sim p_\omega, \pi_\theta}[\text{IS}(\pi_\theta, p_\omega, H)]$ by verifying the boundedness of its second derivative.

$$\frac{\partial^2}{\partial^2 \omega} \mathbb{V}_{H \sim p_\omega, \pi_\theta}[\text{IS}(\pi_\theta, p_\omega, H)]$$

$$= \frac{\partial}{\partial \omega} \mathbb{E}_{H \sim p_\omega, \pi_\theta}\left[\text{IS}(\pi_e, \pi_\theta, H)^2 \sum_{t=0}^{T-1} \log(p_\omega(S_{t+1}|S_t, A_t))\right]$$

$$- 2\mathbb{E}_{H \sim p_\omega, \pi_\theta}[\text{IS}(\pi_e, \pi_\theta, H)]\mathbb{E}_{H \sim p_\omega, \pi_\theta}\left[\text{IS}(\pi_e, \pi_\theta, H) \sum_{t=0}^{T-1} \log(p_\omega(S_{t+1}|S_t, A_t))\right]$$

$$= \frac{\partial}{\partial \omega}\left(\sum_h \left(p(h)m_{p_\omega}(h)\text{IS}(\pi_e, \pi_\theta, H)^2 \sum_{t=0}^{T-1} \frac{\partial}{\partial \omega} \log(p_\omega(S_{t+1}|S_t, A_t))\right)\right.$$

$$- 2\sum_h \left(p(h)m_{p_\omega}(h)\text{IS}(\pi_e, \pi_\theta, H)\right)$$

$$\left. \cdot \sum_h \left(p(h)m_{p_\omega}(h)\text{IS}(\pi_e, \pi_\theta, H) \sum_{t=0}^{T-1} \frac{\partial}{\partial \omega} \log(p_\omega(S_{t+1}|S_t, A_t))\right)\right) \quad \text{(By Lemma 1 and (5))}$$

$$
\begin{aligned}
=& \frac{\partial}{\partial \omega} \left( \sum_h \left( p(h) m_{p_\omega}(h) \mathrm{IS}(\pi_e, \pi_\theta, H)^2 \frac{\partial}{\partial \omega} \log m_{p_\omega}(h) \right) \right. \\
& \left. -2 \sum_h \left( p(h) m_{p_\omega}(h) \mathrm{IS}(\pi_e, \pi_\theta, H) \right) \cdot \sum_h \left( p(h) m_{p_\omega}(h) \mathrm{IS}(\pi_e, \pi_\theta, H) \frac{\partial}{\partial \omega} \log m_{p_\omega}(h) \right) \right) \\
=& \frac{\partial}{\partial \omega} \left( \sum_h \left( p(h) m_{p_\omega}(h) \mathrm{IS}(\pi_e, \pi_\theta, H)^2 \frac{1}{m_{p_\omega}(h)} \frac{\partial}{\partial \omega} m_{p_\omega}(h) \right) \right. \\
& \left. -2 \sum_h \left( p(h) m_{p_\omega}(h) \mathrm{IS}(\pi_e, \pi_\theta, H) \right) \cdot \sum_h \left( p(h) m_{p_\omega}(h) \mathrm{IS}(\pi_e, \pi_\theta, H) \frac{1}{m_{p_\omega}(h)} \frac{\partial}{\partial \omega} m_{p_\omega}(h) \right) \right) \\
=& \frac{\partial}{\partial \omega} \left( \sum_h \left( p(h) \mathrm{IS}(\pi_e, \pi_\theta, H)^2 \frac{\partial}{\partial \omega} m_{p_\omega}(h) \right) \right. \\
& \left. -2 \sum_h \left( p(h) m_{p_\omega}(h) \mathrm{IS}(\pi_e, \pi_\theta, H) \right) \cdot \sum_h \left( p(h) \mathrm{IS}(\pi_e, \pi_\theta, H) \frac{\partial}{\partial \omega} m_{p_\omega}(h) \right) \right) \\
=& \sum_h p(h) \left( \underbrace{\mathrm{IS}(\pi_e, \pi_\theta, H)^2}_{(1)} \underbrace{\frac{\partial^2}{\partial^2 \omega} m_{p_\omega}(h)}_{(2)} \right) \\
& -2 \frac{\partial}{\partial \omega} \left[ \sum_h \left( p(h) m_{p_\omega}(h) \mathrm{IS}(\pi_e, \pi_\theta, H) \right) \cdot \sum_h \left( p(h) \mathrm{IS}(\pi_e, \pi_\theta, H) \frac{\partial}{\partial \omega} m_{p_\omega}(h) \right) \right].
\end{aligned}
$$

We further decompose the term in the square brackets.

$$
\begin{aligned}
& \frac{\partial}{\partial \omega} \left[ \sum_h \left( p(h) m_{p_\omega}(h) \mathrm{IS}(\pi_e, \pi_\theta, H) \right) \cdot \sum_h \left( p(h) \mathrm{IS}(\pi_e, \pi_\theta, H) \frac{\partial}{\partial \omega} m_{p_\omega}(h) \right) \right] \\
=& \sum_h p(h) \frac{\partial}{\partial \omega} (m_{p_\omega}(h) \mathrm{IS}(\pi_e, \pi_\theta, H)) \cdot \sum_h \left( p(h) \mathrm{IS}(\pi_e, \pi_\theta, H) \frac{\partial}{\partial \omega} m_{p_\omega}(h) \right) \\
& + \sum_h \left( p(h) m_{p_\omega}(h) \mathrm{IS}(\pi_e, \pi_\theta, H) \right) \cdot \sum_h p(h) \frac{\partial}{\partial \omega} \left( \mathrm{IS}(\pi_e, \pi_\theta, H) \frac{\partial}{\partial \omega} m_{p_\omega}(h) \right) \\
=& \sum_h p(h) \left( \underbrace{\mathrm{IS}(\pi_e, \pi_\theta, H)}_{(3)} \underbrace{\frac{\partial}{\partial \omega} m_{p_\omega}(h)}_{(4)} \right) \cdot \sum_h p(h) \left( \mathrm{IS}(\pi_e, \pi_\theta, H) \frac{\partial}{\partial \omega} m_{p_\omega}(h) \right) \\
& + \sum_h p(h) \left( \underbrace{m_{p_\omega}(h)}_{(5)} \mathrm{IS}(\pi_e, \pi_\theta, H) \right) \cdot \sum_h p(h) \left( \mathrm{IS}(\pi_e, \pi_\theta, H) \frac{\partial^2}{\partial^2 \omega} m_{p_\omega}(h) \right).
\end{aligned}
$$

Notice that since $p(h)$ is defined as $\frac{Pr(H=h|\pi)}{w_\pi(h)}$ in (4), where $\Pr(H = h)$ is trivially bounded and $w_\pi(h)$ is always positive. Thus, $p(h)$ is bounded. We then analyze the boundedness of $\frac{\partial^2}{\partial^2 \omega} V_{H \sim p_\omega, \pi_\theta}[\mathrm{IS}(\pi_\theta, p_\omega, H)]$ through the above 5 terms.

For (1) and (3), the quotient $\frac{\pi_e(a|s)}{\pi_\theta(a|s)}$ is bounded above by assumption. Besides, since the reward is bounded, so is $g(h)$. Therefore, both (1), $\mathrm{IS}(\pi_e, \pi_\theta, H)^2$ and (3) $\mathrm{IS}(\pi_e, \pi_\theta, H)$ are bounded.

For (5), it is bounded because $m_{p_\omega}(h) = \prod_{t=0}^{T-1} p_\omega(S_{t+1}|S_t, A_t) \leq 1$. Then, for (4),

$$
\begin{aligned}
\frac{\partial}{\partial \omega} m_{p_\omega}(h) =& \frac{\partial}{\partial \omega} \prod_{t=0}^{T-1} p_\omega(S_{t+1}|S_t, A_t) \\
=& \sum_{t=0}^{T-1} \frac{\partial}{\partial \omega} p_\omega(S_{t+1}|S_t, A_t) \frac{\prod_{i=0}^{T-1} p_\omega(S_{i+1}|S_i, A_i)}{p_\omega(S_{t+1}|S_t, A_t)}.
\end{aligned}
$$

Here, $\frac{\partial}{\partial \omega} p_\omega(S_{t+1}|S_t, A_t)$ is bounded by construction and $\frac{\prod_{i=0}^{T-1} p_\omega(S_{i+1}|S_i, A_i)}{p_\omega(S_{t+1}|S_t, A_t)} \leq 1$. Thus, (4) is bounded. Lastly, for (2)

$$\frac{\partial^2}{\partial^2 \omega} m_{p_\omega}(h)$$

$$= \frac{\partial}{\partial \omega} \sum_{t=0}^{T-1} \frac{\partial}{\partial \omega} p_\omega(S_{t+1}|S_t, A_t) \frac{\prod_{i=0}^{T-1} p_\omega(S_{i+1}|S_i, A_i)}{p_\omega(S_{t+1}|S_t, A_t)}$$

$$= \frac{\partial}{\partial \omega} \sum_{t=0}^{T-1} \frac{\partial}{\partial \omega} p_\omega(S_{t+1}|S_t, A_t) \prod_{i \neq t} p_\omega(S_{i+1}|S_i, A_i)$$

$$= \sum_{t=0}^{T-1} \frac{\partial^2}{\partial^2 \omega} p_\omega(S_{t+1}|S_t, A_t) \prod_{i \neq t} p_\omega(S_{i+1}|S_i, A_i) + \frac{\partial}{\partial \omega} p_\omega(S_{t+1}|S_t, A_t)$$

$$\cdot \sum_{i \neq t} \frac{\partial}{\partial \omega} p_\omega(S_{i+1}|S_i, A_i) \prod_{j \neq t,i} p_\omega(S_{j+1}|S_j, A_j),$$

which is bounded because $p_\omega$ is constructed to be twice differentiable with bounded first and second derivatives.

Therefore, we conclude that the gradient objective $\frac{\partial}{\partial \omega} V_{H \sim p_\omega, \pi_\theta}[\text{IS}(\pi_\theta, p_\omega, H)]$ is Lipschitz continuous w.r.t. $\omega$, verifying condition 1.

Finally, we show that the variance of the gradient estimate used by Algorithm 1 is bounded. According to Algorithm 1, we use the unbiased estimate as

$$\frac{\partial}{\partial \omega} V_{H \sim p_\omega, \pi_\theta}[\text{IS}(\pi_\theta, p_\omega, H)] \approx \underbrace{\text{IS}(\pi_\theta, p_\omega, H)^2 \sum_{t=0}^{T-1} \frac{\partial}{\partial \omega} \log(p_\omega(S_{t+1}|S_t, A_t))}_{A}$$

$$\underbrace{- 2\text{IS}(\pi_\theta, p_\omega, H)\text{IS}(\pi_\theta, p_\omega, H) \sum_{t=0}^{T-1} \frac{\partial}{\partial \omega} \log(p_\omega(S_{t+1}|S_t, A_t))}_{B}.$$

Then, the variance of the estimate is decomposed into

$$\mathbb{V}[A] + \mathbb{V}[B] + 2\text{Cov}[A, B],$$

where $\text{Cov}[A, B] \leq \sqrt{\mathbb{V}[A]} \cdot \sqrt{\mathbb{V}[B]}$ by the Cauchy-Schwarz inequality. Thus, it is sufficient to show the boundedness of $\mathbb{V}[A]$ and $\mathbb{V}[B]$. For $\mathbb{V}[A]$, since the variance of a bounded random variable is bounded, we aim to demonstrate that for any trajectory $h$, the term $\text{IS}(\pi_\theta, p_\omega, H)^2 \sum_{t=0}^{T-1} \log(p_\omega(S_{t+1}|S_t, A_t))$ is bounded.

$$\text{IS}(\pi_\theta, p_\omega, H)^2 \sum_{t=0}^{T-1} \frac{\partial}{\partial \omega} \log(p_\omega(S_{t+1}|S_t, A_t))$$

$$= \text{IS}(\pi_\theta, p_\omega, H)^2 \sum_{t=0}^{T-1} \frac{\partial}{\partial \omega} \log(p_\omega(S_{t+1}|S_t, A_t))$$

$$= \text{IS}(\pi_\theta, p_\omega, H)^2 \frac{\partial}{\partial \omega} \log m_{p_\omega}(h)$$

$$= \text{IS}(\pi_\theta, p_\omega, H)^2 \frac{\frac{\partial}{\partial \omega} m_{p_\omega}(h)}{m_{p_\omega}(h)}. \tag{7}$$

The boundedness of $\text{IS}(\pi_\theta, p_\omega, H)^2$ and $\frac{\partial}{\partial \omega} m_{p_\omega}(h)$ is shown by the argument above for term (3) and (4). And the boundedness of $\frac{1}{m_{p_\omega}(h)} = \frac{1}{\prod_{t=0}^{T-1} p_\omega(S_{t+1}|S_t, A_t)}$ comes from the fact that $p_\omega$ is always nonzero by construction. Thus, we conclude that $\mathbb{V}[A]$ is bounded.

Next, we decompose term $B$ into two parts because of the different samples used to estimate them:

$$\underbrace{\text{IS}(\pi_\theta, p_\omega, H)}_{C} \underbrace{\text{IS}(\pi_\theta, p_\omega, H) \sum_{t=0}^{T-1} \frac{\partial}{\partial \omega} \log(p_\omega(S_{t+1}|S_t, A_t))}_{D}.$$

We then have

$$\mathbb{V}[B] = \mathbb{V}[CD] = \mathbb{E}[C^2]\mathbb{V}[D] + \mathbb{E}[D^2]\mathbb{V}[C].$$

We show their boundedness term by term.

$$\mathbb{E}[C^2] = \mathbb{E}_{H\sim p_\omega, \pi_\theta}[\text{IS}(\pi_\theta, p_\omega, H)^2] = \sum_h p(h)m_{p_\omega}(h)\text{IS}(\pi_e, \pi_\theta, H)^2, \tag{8}$$

where each term is shown to be bounded above. Next, by the derivation from (7),

$$\mathbb{E}[D^2] = \sum_h p(h)m_{p_\omega}(h)\text{IS}(\pi_e, \pi_\theta, H)^2 \left(\frac{\frac{\partial}{\partial\omega}m_{p_\omega}(h)}{m_{p_\omega}(h)}\right)^2,$$

where the boundedness follows from the analysis of (8) and (7).

As for the two variance terms, $\mathbb{V}[C]$ and $\mathbb{V}[D]$, we show the boundedness of the random variable $C$ and $D$ for each trajectory $h$, where $\text{IS}(\pi_\theta, p_\omega, H)$ is shown to be bounded in term (3) above, and the boundedness of $\sum_{t=0}^{T-1} \frac{\partial}{\partial\omega} \log(p_\omega(S_{t+1}|S_t, A_t))$ is incorporated in (7).

Therefore, we conclude that the variance of our estimate is bounded. By far, we show that the three conditions of Proposition 3 in Bertsekas and Tsitsiklis (2000) are satisfied, demonstrating the convergence of Algorithm 1.

$\square$

### A.4 PROOF OF THEOREM 2

*Proof.*

**Theorem 2** (Transition Gradient of Variance with KL). *For a fixed behavior policy $\pi_\theta$ and a regularization coefficient $\eta > 0$,*

$$\frac{\partial}{\partial\omega}\mathbb{V}_{H\sim p_\omega, \pi_\theta}[\text{OPE}(\pi_e, \pi_\theta, H)] - \eta D_{\text{KL}}(\Pr(H|p_\omega)\| \Pr(H|p_{\omega_0}))$$

$$=\mathbb{E}_{H\sim p_\omega, \pi_\theta}\left[\text{OPE}(\pi_e, \pi_\theta, H)^2\frac{\partial}{\partial\omega}\ell_{p_\omega}\right]$$

$$- 2\mathbb{E}_{H\sim p_\omega, \pi_\theta}[\text{OPE}(\pi_e, \pi_\theta, H)]\mathbb{E}_{H\sim p_\omega, \pi_\theta}\left[\text{OPE}(\pi_e, \pi_\theta, H)\frac{\partial}{\partial\omega}\ell_{p_\omega}\right]$$

$$- \eta\mathbb{E}_{H\sim p_\omega, \pi_\theta}\left[\left(\frac{\partial}{\partial\omega}\ell_{p_\omega}\right)\left(1 + \ell_{p_\omega} - \ell_{p_{\omega_0}}\right)\right].$$

We begin by manipulating the KL-divergence term.

$$D_{\text{KL}}(\Pr(H|p_\omega)\| \Pr(H|p_{\omega_0})) =\mathbb{E}_{H\sim p_\omega, \pi_\theta}\left[\log\frac{\Pr(H|p_\omega)}{\Pr(H|p_{\omega_0})}\right]$$

$$=\mathbb{E}_{H\sim p_\omega, \pi_\theta}\left[\log\frac{m_{p_\omega}(H)}{m_{p_{\omega_0}}(H)}\right] \qquad \text{(By (5))}$$

$$=\mathbb{E}_{H\sim p_\omega, \pi_\theta}\left[\log m_{p_\omega}(H) - \log m_{p_{\omega_0}}(H)\right].$$

Next, we decompose the following gradient:

$$\frac{\partial}{\partial\omega}\log m_{p_\omega}(H) \tag{9}$$

$$=\sum_{t=0}^{T-1}\frac{\partial}{\partial\omega}\log p_\omega(S_{t+1}|S_t, A_t)$$

$$=\sum_{t=0}^{T-1}\frac{\partial}{\partial\omega}\log(p_\omega(S_{t+1}|S_t, A_t)). \qquad \text{(By definition)}$$

Then, we take the gradient of the KL-divergence with respect to $\omega$:

$$\frac{\partial}{\partial\omega}D_{\mathrm{KL}}(\Pr(H|p_\omega)\|\Pr(H|p_{\omega_0})) \tag{10}$$

$$=\frac{\partial}{\partial\omega}\mathbb{E}_{H\sim p_\omega,\pi_\theta}\big[\log m_{p_\omega}(H)-\log m_{p_{\omega_0}}(H)\big]$$

$$=\frac{\partial}{\partial\omega}\sum_h \Pr(H=h|p_\omega)\big[\log m_{p_\omega}(h)-\log m_{p_{\omega_0}}(h)\big]$$

$$=\frac{\partial}{\partial\omega}\sum_h p(h)m_{p_\omega}(h)\big[\log m_{p_\omega}(h)-\log m_{p_{\omega_0}}(h)\big] \qquad \text{(By (5))}$$

$$=\sum_h p(h)\Big[\frac{\partial}{\partial\omega}m_{p_\omega}(h)\log m_{p_\omega}(h)-\log m_{p_{\omega_0}}(h)\frac{\partial}{\partial\omega}m_{p_\omega}(h)\Big]$$

$$=\sum_h p(h)\Big[\log m_{p_\omega}(h)\frac{\partial}{\partial\omega}m_{p_\omega}(h)+m_{p_\omega}(h)\frac{\partial}{\partial\omega}\log m_{p_\omega}(h)$$

$$-\log m_{p_{\omega_0}}(h)m_{p_\omega}(h)\sum_{t=0}^{T-1}\frac{\partial}{\partial\omega}\log(p_\omega(S_{t+1}|S_t,A_t))\Big] \qquad \text{(By (6))}$$

$$=\sum_h p(h)\Big[\log m_{p_\omega}(h)m_{p_\omega}(h)\sum_{t=0}^{T-1}\frac{\partial}{\partial\omega}\log(p_\omega(S_{t+1}|S_t,A_t))+m_{p_\omega}(h)\sum_{t=0}^{T-1}\frac{\partial}{\partial\omega}\log(p_\omega(S_{t+1}|S_t,A_t))$$

$$-\log m_{p_{\omega_0}}(h)m_{p_\omega}(h)\sum_{t=0}^{T-1}\frac{\partial}{\partial\omega}\log(p_\omega(S_{t+1}|S_t,A_t))\Big] \qquad \text{(By (6) (9))}$$

$$=\sum_h p(h)m_{p_\omega}(h)\sum_{t=0}^{T-1}\frac{\partial}{\partial\omega}\log(p_\omega(S_{t+1}|S_t,A_t))\big[\log m_{p_\omega}(h)+1-\log m_{p_{\omega_0}}(h)\big]$$

$$=\sum_h \Pr(H=h|p_\omega)\sum_{t=0}^{T-1}\Big[\frac{\partial}{\partial\omega}\log(p_\omega(S_{t+1}|S_t,A_t))\Big]\big[\log m_{p_\omega}(h)+1-\log m_{p_{\omega_0}}(h)\big] \quad \text{(By (5))}$$

$$=\mathbb{E}_{H\sim p_\omega,\pi_\theta}\Big[\Big(\frac{\partial}{\partial\omega}\ell_{p_\omega}\Big)\big(1+\ell_{p_\omega}-\ell_{p_{\omega_0}}\big)\Big]. \qquad \text{(By (3))}$$

Thus,

$$\frac{\partial}{\partial\omega}\mathbb{V}_{H\sim p_\omega,\pi_\theta}\left[\mathrm{OPE}(\pi_e,\pi_\theta,H)\right]-\eta D_{\mathrm{KL}}(\Pr(H|p_\omega)\|\Pr(H|p_{\omega_0}))$$

$$=\mathbb{E}_{H\sim p_\omega,\pi_\theta}\Big[\mathrm{OPE}(\pi_e,\pi_\theta,H)^2\sum_{t=0}^{T-1}\log(p_\omega(S_{t+1}|S_t,A_t))\Big]-2\mathbb{E}_{H\sim p_\omega,\pi_\theta}[\mathrm{OPE}(\pi_e,\pi_\theta,H)]$$

$$\cdot\,\mathbb{E}_{H\sim p_\omega,\pi_\theta}\Big[\mathrm{OPE}(\pi_e,\pi_\theta,H)\sum_{t=0}^{T-1}\log(p_\omega(S_{t+1}|S_t,A_t))\Big]-\eta\frac{\partial}{\partial\omega}D_{\mathrm{KL}}(\Pr(H|p_\omega)\|\Pr(H|p_{\omega_0}))$$
$$\text{(By Lemma 1)}$$

$$=\mathbb{E}_{H\sim p_\omega,\pi_\theta}\Big[\mathrm{OPE}(\pi_e,\pi_\theta,H)^2\frac{\partial}{\partial\omega}\ell_{p_\omega}\Big]-2\mathbb{E}_{H\sim p_\omega,\pi_\theta}[\mathrm{OPE}(\pi_e,\pi_\theta,H)]\mathbb{E}_{H\sim p_\omega,\pi_\theta}\Big[\mathrm{OPE}(\pi_e,\pi_\theta,H)\frac{\partial}{\partial\omega}\ell_{p_\omega}\Big]$$

$$-\eta\mathbb{E}_{H\sim p_\omega,\pi_\theta}\Big[\Big(\frac{\partial}{\partial\omega}\ell_{p_\omega}\Big)\big(1+\ell_{p_\omega}-\ell_{p_{\omega_0}}\big)\Big]. \qquad \text{(By (10))}$$

$\square$

## A.5 PROOF OF THEOREM 3

**Theorem 3** (Off-Transition Gradient of Variance). *When $p_\omega \neq p_{\omega_0}$, for a fixed behavior policy $\pi_\theta$,*

$$\frac{\partial}{\partial\omega}\mathbb{V}_{H\sim p_{\omega_0},\pi_\theta}[\text{OPE}(\pi_e,\pi_\theta,p_\omega,H)]$$
$$=2\mathbb{E}_{H\sim p_{\omega_0},\pi_\theta}\big[\text{OPE}^2(\pi_e,\pi_\theta,p_\omega,H)\tfrac{\partial}{\partial\omega}\ell_{p_\omega}\big]$$
$$-2\mathbb{E}_{H\sim p_{\omega_0},\pi_\theta}[\text{OPE}(\pi_e,\pi_\theta,p_\omega,H)]\cdot\mathbb{E}_{H\sim p_{\omega_0},\pi_\theta}\big[\text{OPE}(\pi_e,\pi_\theta,p_\omega,H)\tfrac{\partial}{\partial\omega}\ell_{p_\omega}\big].$$

*Proof.* For simplification, we define $w_\pi(h)\doteq\prod_{t=0}^{T-1}\pi(A_t|S_t)$ under trajectory $h$. Then,

$$\frac{\partial}{\partial\omega}\mathbb{V}_{H\sim p_{\omega_0},\pi_\theta}[\text{OPE}(\pi_e,\pi_\theta,p_\omega,H)]$$

$$=\frac{\partial}{\partial\omega}\big(\mathbb{E}_{H\sim p_{\omega'}}[\text{OPE}^2(\pi_e,\pi_\theta,p_\omega,H)]-\mathbb{E}_{H\sim p_{\omega'}}[\text{OPE}(\pi_e,\pi_\theta,p_\omega,H)]^2\big)$$

$$=\frac{\partial}{\partial\omega}\left(\mathbb{E}_{H\sim p_{\omega'}}\left[\frac{m_{p_\omega}^2(H)}{m_{p_{\omega_0}}^2(H)}\text{OPE}^2(\pi_e,\pi_\theta,H)\right]-\mathbb{E}_{H\sim p_{\omega'}}\left[\frac{m_{p_\omega}(H)}{m_{p_{\omega_0}}(H)}\text{OPE}(\pi_e,\pi_\theta,H)\right]^2\right)$$

$$=\frac{\partial}{\partial\omega}\sum_h\left(\Pr(H=h|p_{\omega_0})\frac{m_{p_\omega}^2(H)}{m_{p_{\omega_0}}^2(H)}\text{OPE}^2(\pi_e,\pi_\theta,H)\right)-2\mathbb{E}_{H\sim p_{\omega'}}[\text{OPE}(\pi_e,\pi_\theta,p_\omega,H)]$$

$$\frac{\partial}{\partial\omega}\mathbb{E}_{H\sim p_{\omega'}}\left[\frac{m_{p_\omega}(H)}{m_{p_{\omega_0}}(H)}\text{OPE}(\pi_e,\pi_\theta,H)\right]$$

$$=\sum_h\left(p(h)\frac{1}{m_{p_{\omega_0}}(H)}\text{OPE}^2(\pi_e,\pi_\theta,H)\frac{\partial}{\partial\omega}m_{p_\omega}^2(h)\right)$$

$$-2\mathbb{E}_{H\sim p_{\omega'}}[\text{OPE}(\pi_e,\pi_\theta,p_\omega,H)]\frac{\partial}{\partial\omega}\sum_h\left(p(h)m_{p_{\omega_0}(h)}\frac{m_{p_\omega}(h)}{m_{p_{\omega_0}}(h)}\text{OPE}(\pi_e,\pi_\theta,H)\right)\quad\text{(By (5))}$$

$$=2\sum_h\left(p(h)\frac{m_{p_\omega}(h)}{m_{p_{\omega_0}}(h)}\text{OPE}^2(\pi_e,\pi_\theta,H)\frac{\partial}{\partial\omega}m_{p_\omega}(h)\right)$$

$$-2\mathbb{E}_{H\sim p_{\omega'}}[\text{OPE}(\pi_e,\pi_\theta,p_\omega,H)]\sum_h\left(p(h)\text{OPE}(\pi_e,\pi_\theta,H)\frac{\partial}{\partial\omega}m_{p_\omega}(h)\right)\quad\text{(By (5))}$$

$$=2\sum_h\left(p(h)\text{OPE}^2(\pi_e,\pi_\theta,H)\frac{m_{p_\omega}(h)}{m_{p_{\omega_0}}(h)}m_{p_\omega}(h)\frac{\partial}{\partial\omega}\ell_{p_\omega}\right)$$

$$-2\mathbb{E}_{H\sim p_{\omega'}}[\text{OPE}(\pi_e,\pi_\theta,p_\omega,H)]\sum_h\left(p(h)\text{OPE}(\pi_e,\pi_\theta,H)m_{p_\omega}(h)\frac{\partial}{\partial\omega}\ell_{p_\omega}\right)\quad\text{(By (6))}$$

$$=2\sum_h\left(p(h)m_{p_{\omega_0}}(h)\frac{m_{p_\omega}^2(h)}{m_{p_{\omega_0}}^2(h)}\text{OPE}^2(\pi_e,\pi_\theta,H)\frac{\partial}{\partial\omega}\ell_{p_\omega}\right)$$

$$-2\mathbb{E}_{H\sim p_{\omega'}}[\text{OPE}(\pi_e,\pi_\theta,p_\omega,H)]$$

$$\cdot\sum_h\left(p(h)m_{p_{\omega_0}}(h)\frac{m_{p_\omega}(h)}{m_{p_{\omega_0}}(h)}\text{OPE}(\pi_e,\pi_\theta,H)\sum_{t=0}^{T-1}\log(p_\omega(S_{t+1}|S_t,A_t))\right)\quad\text{(By (6))}$$

$$=2\sum_h\left(\Pr(H=h|p_{\omega_0})\text{OPE}^2(\pi_e,\pi_\theta,p_\omega,H)\frac{\partial}{\partial\omega}\ell_{p_\omega}\right)$$

$$-2\mathbb{E}_{H\sim p_{\omega'}}[\text{OPE}(\pi_e,\pi_\theta,p_\omega,H)]$$

$$\cdot \sum_h \left( \Pr(H = h | p_{\omega_0}) \text{OPE}(\pi_e, \pi_\theta, p_\omega, H) \sum_{t=0}^{T-1} \frac{\partial}{\partial \omega} \log(p_\omega(S_{t+1} | S_t, A_t)) \right) \quad \text{(By (5))}$$

$$=2\mathbb{E}_{H \sim p_{\omega'}} \left[ \text{OPE}^2(\pi_e, \pi_\theta, p_\omega, H) \frac{\partial}{\partial \omega} \ell_{p_\omega} \right]$$

$$- 2\mathbb{E}_{H \sim p_{\omega'}}[\text{OPE}(\pi_e, \pi_\theta, p_\omega, H)]\mathbb{E}_{H \sim p_{\omega'}} \left[ \text{OPE}(\pi_e, \pi_\theta, p_\omega, H) \frac{\partial}{\partial \omega} \ell_{p_\omega} \right].$$

□

### A.6  PROOF OF THEOREM 4

**Theorem 4** (Off-transition Gradient of Variance with KL). *For a fixed behavior policy $\pi_\theta$ and a regularization coefficient $\eta > 0$,*

$$\frac{\partial}{\partial \omega} \mathbb{V}_{H \sim p_{\omega_0}, \pi_\theta}[\text{OPE}(\pi_e, \pi_\theta, p_\omega, H)] - \eta D_{\text{KL}}(\Pr(H | p_{\omega_0}) \| \Pr(H | p_\omega))$$

$$=2\mathbb{E}_{H \sim p_{\omega_0}, \pi_\theta} \left[ \text{OPE}^2(\pi_e, \pi_\theta, p_\omega, H) \frac{\partial}{\partial \omega} \ell_{p_\omega} \right]$$

$$- 2\mathbb{E}_{H \sim p_{\omega_0}, \pi_\theta}[\text{OPE}(\pi_e, \pi_\theta, p_\omega, H)]\mathbb{E}_{H \sim p_{\omega_0}, \pi_\theta} \left[ \text{OPE}(\pi_e, \pi_\theta, p_\omega, H) \frac{\partial}{\partial \omega} \ell_{p_\omega} \right]$$

$$- \eta \mathbb{E}_{H \sim p_{\omega_0}, \pi_\theta} \left[ -\frac{\partial}{\partial \omega} \ell_{p_\omega} \right].$$

The KL-divergence between two probability distribution $p$ and $q$ is defined as $D_{\text{KL}}(p \| q) \doteq \mathbb{E}_{X \sim p} \left[ \log \frac{p(X)}{q(X)} \right]$. Therefore, the KL-divergence between the trajectory distribution of the target transition $p_\omega$ and the simulator's transition $p_{\omega_0}$ is given by

$$D_{\text{KL}}(\Pr(H | p_{\omega_0}) \| \Pr(H | p_\omega)) = \mathbb{E}_{H \sim p_{\omega_0}, \pi_\theta} \left[ \log \frac{\Pr(H | p_{\omega_0})}{\Pr(H | p_\omega)} \right]$$

$$= \mathbb{E}_{H \sim p_{\omega_0}, \pi_\theta} \left[ \log \frac{m_{p_{\omega_0}}(H)}{m_{p_\omega}(H)} \right] \quad \text{(By (5))}$$

$$= \mathbb{E}_{H \sim p_{\omega_0}, \pi_\theta} \left[ \log m_{p_{\omega_0}}(H) - \log m_{p_\omega}(H) \right].$$

We take the gradient of the KL-divergence with respect to $\omega$:

$$\frac{\partial}{\partial \omega} D_{\text{KL}}(\Pr(H | p_{\omega_0}) \| \Pr(H | p_\omega)) = \frac{\partial}{\partial \omega} \mathbb{E}_{H \sim p_{\omega_0}, \pi_\theta} \left[ \log m_{p_{\omega_0}}(H) - \log m_{p_\omega}(H) \right]$$

$$= \mathbb{E}_{H \sim p_{\omega_0}, \pi_\theta} \left[ -\frac{\partial}{\partial \omega} \log m_{p_\omega}(H) \right]$$

$$= \mathbb{E}_{H \sim p_{\omega_0}, \pi_\theta} \left[ -\sum_{t=0}^{T-1} \frac{\partial}{\partial \omega} \log p_\omega(S_{t+1} | S_t, A_t) \right]$$

$$= \mathbb{E}_{H \sim p_{\omega_0}, \pi_\theta} \left[ -\sum_{t=0}^{T-1} \frac{\partial}{\partial \omega} \log(p_\omega(S_{t+1} | S_t, A_t)) \right]. \quad (11)$$

Thus,

$$
\begin{aligned}
&\frac{\partial}{\partial \omega} \mathbb{V}_{H \sim p_{\omega_0}, \pi_\theta}[\mathrm{OPE}(\pi_e, \pi_\theta, p_\omega, H)] - \eta D_{\mathrm{KL}}(\Pr(H|p_{\omega_0}) \| \Pr(H|p_\omega)) \\
&= 2\mathbb{E}_{H \sim p_{\omega'}} \left[ \mathrm{OPE}^2(\pi_e, \pi_\theta, p_\omega, H) \tfrac{\partial}{\partial \omega} \ell_{p_\omega} \right] \\
&\quad - 2\mathbb{E}_{H \sim p_{\omega'}}[\mathrm{OPE}(\pi_e, \pi_\theta, p_\omega, H)] \mathbb{E}_{H \sim p_{\omega'}} \left[ \mathrm{OPE}(\pi_e, \pi_\theta, p_\omega, H) \tfrac{\partial}{\partial \omega} \ell_{p_\omega} \right] \\
&\quad - \tfrac{\partial}{\partial \omega} \eta D_{\mathrm{KL}}(\Pr(H|p_{\omega_0}) \| \Pr(H|p_\omega)) \qquad\qquad\qquad \text{(By Lemma 4)} \\
&= 2\mathbb{E}_{H \sim p_{\omega'}} \left[ \mathrm{OPE}^2(\pi_e, \pi_\theta, p_\omega, H) \tfrac{\partial}{\partial \omega} \ell_{p_\omega} \right] \\
&\quad - 2\mathbb{E}_{H \sim p_{\omega'}}[\mathrm{OPE}(\pi_e, \pi_\theta, p_\omega, H)] \mathbb{E}_{H \sim p_{\omega'}} \left[ \mathrm{OPE}(\pi_e, \pi_\theta, p_\omega, H) \tfrac{\partial}{\partial \omega} \ell_{p_\omega} \right] \\
&\quad - \eta \mathbb{E}_{H \sim p_{\omega_0}, \pi_\theta} \left[ -\sum_{t=0}^{T-1} \tfrac{\partial}{\partial \omega} \log(p_\omega(S_{t+1}|S_t, A_t)) \right]. \qquad\qquad \text{(By (11))}
\end{aligned}
$$

### A.7  PROOF OF LEMMA 3

*Proof.* By Lemma 2,

$$
\frac{\partial}{\partial \theta} \mathbb{V}_{H \sim p_\omega, \pi_\theta}[\mathrm{IS}(\pi_e, \pi_\theta, H)] = \mathbb{E}_{H \sim p_\omega, \pi_\theta} \left[ -\mathrm{IS}(\pi_e, \pi_\theta, H)^2 \sum_{t=0}^{T-1} \frac{\partial}{\partial \theta} \log \pi_\theta(A_t|S_t) \right]. \quad (12)
$$

To prove the Lipschitz property, we bound each term in the RHS. First, we aim to bound $\left\| \frac{\partial}{\partial \theta} \log \pi_\theta(A_t|S_t) \right\|$. Remember that we define

$$
\pi_\theta(a|s) \doteq \frac{\exp\left(\theta_a^\top \phi(s)\right)}{\sum_{a' \in \mathcal{A}} \exp\left(\theta_{a'}^\top \phi(s)\right)},
$$

where we assumed the linear features $\|\phi(s)\|$ to be bounded by a constant $B$. Here, $\theta = \{\theta_a\}_{a \in \mathcal{A}}$ is the whole parameter matrix, and $\theta_a$ is the column for action $a$ specifically. From Wang et al. (2023a), we know that

$$
\left\| \frac{\partial}{\partial \theta} \log \pi_\theta(a|s) \right\|^2 = \sum_{a' \in \mathcal{A}} \left\| \frac{\partial}{\partial \theta_{a'}} \log \pi_\theta(a|s) \right\|^2.
$$

Further decomposing, we get

$$
\begin{aligned}
\left\| \frac{\partial}{\partial \theta} \log \pi_\theta(a|s) \right\| &= \left[ \|\phi(s)\|_2^2 \left( 1 - 2\pi_\theta(a|s) + \sum_{a' \in \mathcal{A}} \pi_\theta(a'|s)^2 \right) \right]^{\frac{1}{2}} \\
&\leq \sqrt{2}B.
\end{aligned}
$$

Thus, we have

$$
\left\| \sum_{t=0}^{T-1} \frac{\partial}{\partial \theta} \log \pi_\theta(A_t|S_t) \right\| \leq \sqrt{2}BT. \quad (13)
$$

We also make the standard assumption that the quotient $\frac{\pi_e(a|s)}{\pi_\theta(a|s)}$ is bounded above by a constant $C$ for all $s$, $a$, and $\theta$.

$$
\left\| \mathrm{IS}(\pi_e, \pi_\theta, H)^2 \right\| = \left\| \left( \frac{\prod_{t=0}^{T-1} \pi_e(A_t|S_t)}{\prod_{t=0}^{T-1} \pi_\theta(A_t|S_t)} g(H) \right)^2 \right\| \leq C^{2T} T^2, \quad (14)
$$

since we assume the reward is bounded above by 1.

Then,

$$
\left\| \frac{\partial}{\partial \theta} V_{H \sim p_\omega, \pi_\theta}[\mathrm{IS}(\pi_e, \pi_\theta, H)] \right\| \leq \sqrt{2}BC^{2T} T^3.
$$

Thus, the objective function $\mathbb{V}_{H \sim p_\omega, \pi_\theta}[\mathrm{IS}(\pi_e, \pi_\theta, H)]$ is $L_\Theta$-Lipschitz in $\theta$ with $L_\Theta = \sqrt{2}BC^{2T} T^3$.

Next, we aim to show that the objective function $\mathbb{V}_{H \sim p_\omega, \pi_\theta}[\mathrm{IS}(\pi_e, \pi_\theta, H)]$ is $\ell_\Theta$-smooth in $\theta$.

Under trajectory $h$, define $w_{\pi_\theta}(h) \doteq \prod_{t=0}^{T-1} \pi_\theta(A_t, S_t)$ , and $\tilde{p}(h) = \frac{\mathrm{Pr}(H=h|\pi_\theta)}{w_{\pi_\theta}(h)}$. For a fixed transition $p_\omega$, we have the following decomposition as also shown in Hanna et al. (2024):

$$
\frac{\partial^2}{\partial \theta^2} \mathbb{V}_{H \sim p_\omega, \pi_\theta}[\mathrm{IS}(\pi_\theta, p_\omega, H)]
$$

$$
= \frac{\partial}{\partial \theta} \mathbb{E}_{H \sim p_\omega, \pi_\theta} \left[ -\mathrm{IS}(\pi_e, \pi_\theta, H)^2 \sum_{t=0}^{T-1} \frac{\partial}{\partial \theta} \log \pi_\theta(A_t|S_t) \right] \qquad \text{(By (12))}
$$

$$
= \frac{\partial}{\partial \theta} \sum_h \tilde{p}(h) w_{\pi_\theta}(h) \left( -\mathrm{IS}(\pi_e, \pi_\theta, H)^2 \frac{\partial}{\partial \theta} w_{\pi_\theta}(h) \right)
$$

$$
= \frac{\partial}{\partial \theta} \sum_h \tilde{p}(h) w_{\pi_\theta}(h) \left( -\mathrm{IS}(\pi_e, \pi_\theta, H)^2 \frac{\partial}{\partial \theta} w_{\pi_\theta}(h) \frac{1}{w_{\pi_\theta}(h)} \right)
$$

$$
= \frac{\partial}{\partial \theta} \sum_h -\tilde{p}(h) \mathrm{IS}(\pi_e, \pi_\theta, H)^2 \frac{\partial}{\partial \theta} w_{\pi_\theta}(h)
$$

$$
= \sum_h -\tilde{p}(h) \left[ \frac{\partial}{\partial \theta} \mathrm{IS}(\pi_e, \pi_\theta, H)^2 \frac{\partial}{\partial \theta} w_{\pi_\theta}(h) + \mathrm{IS}(\pi_e, \pi_\theta, H)^2 \frac{\partial^2}{\partial^2 \theta} w_{\pi_\theta}(h) \right]. \qquad (15)
$$

For the terms here, we have

$$
\frac{\partial}{\partial \theta} \mathrm{IS}(\pi_e, \pi_\theta, H)^2 = \frac{-2g(h)^2 w_{\pi_e}(h)^2}{w_{\pi_\theta}(h)^3} \frac{\partial}{\partial \theta} w_{\pi_\theta}(h),
$$

$$
\frac{\partial}{\partial \theta} w_{\pi_\theta}(h) = \sum_{t=0}^{T-1} \frac{\partial}{\partial \theta} \pi_\theta(A_t|S_t) \prod_{t'=0, t' \neq t}^{T-1} \pi_\theta(A_{t'}|S_{t'}),
$$

and

$$
\frac{\partial^2}{\partial^2 \theta} w_{\pi_\theta}(h) = \frac{\partial}{\partial \theta} \sum_{t=0}^{T-1} \left( \frac{\partial}{\partial \theta} \pi_\theta(A_t|S_t) \prod_{t'=0, t' \neq t}^{T-1} \pi_\theta(A_{t'}|S_{t'}) \right)
$$

$$
= \sum_{t=0}^{T-1} \left( \frac{\partial^2}{\partial^2 \theta} \pi_\theta(A_t|S_t) \prod_{t \neq t'} \pi_\theta(A_{t'}|S_{t'}) + \frac{\partial}{\partial \theta} \pi_\theta(A_t|S_t) \sum_{t \neq t'} \frac{\partial}{\partial \theta} \pi_\theta(A_{t'}|S_{t'}) \prod_{t'' \neq t, t'} \pi_\theta(A_{t''}|S_{t''}) \right). \tag{16}
$$

Denote $\theta_\alpha = \theta + \alpha u$, where $\alpha \in \mathbb{R}$, $u \in \mathbb{R}^{d|\mathcal{A}|}$, with $d$ being the linear feature dimension. By chain rule, we have, for a fixed transition $p_\omega$,

$$
\frac{\partial^2}{\partial^2 \alpha} \mathbb{V}_{H \sim \pi_{\theta_\alpha}, p_\omega}[\mathrm{IS}(\pi_e, \pi_{\theta_\alpha}, H)] \Big|_{\alpha=0}
$$

$$
= u^\top \frac{\partial^2}{\partial^2 \theta} \mathbb{V}_{H \sim \pi_\theta, p_\omega}[\mathrm{IS}(\pi_e, \pi_\theta, H)] u
$$

$$
= u^\top \sum_h -\tilde{p}(h) \left[ \frac{\partial}{\partial \theta} \mathrm{IS}(\pi_e, \pi_\theta, H)^2 \frac{\partial}{\partial \theta} w_{\pi_\theta}(h)^\top + \mathrm{IS}(\pi_e, \pi_\theta, H)^2 \frac{\partial^2}{\partial^2 \theta} w_{\pi_\theta}(h) \right] u \qquad \text{(By (15))}
$$

$$
= \sum_h -\tilde{p}(h) \left[ \left\langle \frac{\partial}{\partial \theta} \mathrm{IS}(\pi_e, \pi_\theta, H)^2, u \right\rangle \left\langle \frac{\partial}{\partial \theta} w_{\pi_\theta}(h), u \right\rangle + \mathrm{IS}(\pi_e, \pi_\theta, H)^2 u^\top \frac{\partial^2}{\partial^2 \theta} w_{\pi_\theta}(h) u \right]. (17)
$$

We analyze the bound term by term. First, for $\left\langle \frac{\partial}{\partial\theta} \mathrm{IS}(\pi_e, \pi_\theta, H)^2, u \right\rangle$, note that

$$\left\| \frac{\partial}{\partial\theta} \mathrm{IS}(\pi_e, \pi_\theta, H)^2 \right\|$$

$$= \left\| \frac{2g(h)^2 w_{\pi_e}(h)^2}{w_{\pi_\theta}(h)^3} \frac{\partial}{\partial\theta} w_{\pi_\theta}(h) \right\|$$

$$\leq 2T^2 \left\| \frac{w_{\pi_e}(h)^2}{w_{\pi_\theta}(h)^3} w_{\pi_\theta}(h) \sum_{t=0}^{T-1} \frac{\partial}{\partial\theta} \log(\pi_\theta(A_t|S_t)) \right\| \qquad \text{(By (21) of Hanna et al. (2024))}$$

$$\leq 2T^2 C^{2T} \left\| \sum_{t=0}^{T-1} \frac{\partial}{\partial\theta} \log(\pi_\theta(A_t|S_t)) \right\|$$

$$\leq 2\sqrt{2} B T^3 C^{2T}. \qquad\qquad\qquad\qquad\qquad\qquad\qquad\qquad \text{(By (13))}$$

Thus,

$$\left| \left\langle \frac{\partial}{\partial\theta} \mathrm{IS}(\pi_e, \pi_\theta, H)^2, u \right\rangle \right| \leq 2\sqrt{2} B T^3 C^{2T} \|u\|_2. \tag{18}$$

Next, for $\left\langle \frac{\partial}{\partial\theta} w_{\pi_\theta}(h), u \right\rangle$, recall that

$$\left\| \frac{\partial}{\partial\theta} w_{\pi_\theta}(h) \right\|$$

$$= \left\| w_{\pi_\theta}(h) \sum_{t=0}^{T-1} \frac{\partial}{\partial\theta} \log(\pi_\theta(A_t|S_t)) \right\|$$

$$\leq 1 \cdot \sqrt{2} B T.$$

Thus,

$$\left| \left\langle \frac{\partial}{\partial\theta} w_{\pi_\theta}(h), u \right\rangle \right| \leq \sqrt{2} B T \|u\|_2. \tag{19}$$

As for the second term $\mathrm{IS}(\pi_e, \pi_\theta, H)^2 u^\top \frac{\partial^2}{\partial^2\theta} w_{\pi_\theta}(h) u$, remember that by (14),

$$\left\| \mathrm{IS}(\pi_e, \pi_\theta, H)^2 \right\| \leq C^{2T} T^2.$$

Besides, by (16),

$$u^\top \frac{\partial^2}{\partial^2\theta} w_{\pi_\theta}(h) u = \underbrace{\sum_{t=0}^{T-1} u^\top \frac{\partial^2}{\partial^2\theta} \pi_\theta(A_t|S_t) u \prod_{t \neq t'} \pi_\theta(A_{t'}|S_{t'})}_{(a)}$$

$$+ \underbrace{\sum_{t=0}^{T-1} \left\langle \frac{\partial}{\partial\theta} \pi_\theta(A_t|S_t), u \right\rangle \cdot \sum_{t \neq t'} \left\langle \frac{\partial}{\partial\theta} \pi_\theta(A_{t'}|S_{t'}), u \right\rangle \prod_{t'' \neq t, t'} \pi_\theta(A_{t''}|S_{t''})}_{(b)} \tag{20}$$

To bound term (a) and (b), notice that

$$\frac{\partial \pi_\theta(a|s)}{\partial \theta'_a} = \pi_\theta(a|s)(\mathbf{1}\{a' = a\} - \pi_\theta(a'|s))\phi(s),$$

where $\mathbf{1}$ is the indicator function. Recall that we denote. Now we also define a state-wise logit direction $v_s \in \mathbb{R}^{|\mathcal{A}|}$ with each component $v_s(a') \doteq \langle u_{a'}, \phi(s) \rangle$.

Then, for the first derivative,

$$\left|\frac{\partial \pi_{\theta_\alpha}(a|s)}{\partial \alpha}\Big|_{\alpha=0}\right| = \left|\left\langle\frac{\partial \pi_\theta(a|s)}{\partial \theta}, u\right\rangle\right|$$

$$= \left|\pi_\theta(a|s) \cdot \left(v_s(a) - \sum_{a'} \pi_\theta(a'|s)v_s(a')\right)\right|$$

$$\leq \pi_\theta(a|s)\left(|v_s(a)| + \left|\sum_{a'} \pi_\theta(a'|s)v_s(a')\right|\right) \qquad \text{(Triangular Inequality)}$$

$$\leq 2\pi_\theta(a|s)\|v_s\|_2$$

$$\leq 2\|v_s\|_2$$

$$\leq 2\|\phi(s)\|_2\|u\|_2$$

$$\leq 2B\|u\|_2, \tag{21}$$

where $\|u\|_F^2 = \sum_{a\in\mathcal{A}}\|u_a\|_2^2$. Similarly, for the second derivative,

$$\left|\frac{\partial^2 \pi_{\theta_\alpha}(a|s)}{\partial \alpha^2}\Big|_{\alpha=0}\right| = \left|\left\langle\frac{\partial^2 \pi_\theta(a|s)}{\partial \theta^2}v_s, v_s\right\rangle\right|$$

$$= \left|\pi_\theta(a\mid s)\left[(1-\pi_\theta(a\mid s))v_s(a)^2 - \sum_{a'\neq a}\pi_\theta(a'\mid s)\big(v_s(a)-v_s(a')\big)^2 + \sum_{a'}\pi_\theta(a'\mid s)^2 v_s(a')^2\right]\right|$$

$$\leq 5\|v_s\|_2^2$$

$$\leq 5B^2\|u\|_2^2 \tag{22}$$

Now, getting back to (a) in (20), we have

$$|(a)| = \left|\sum_{t=0}^{T-1} u^\top \frac{\partial^2}{\partial^2\theta}\pi_\theta(A_t|S_t)u \prod_{t\neq t'}\pi_\theta(A_{t'}|S_{t'})\right|$$

$$\leq \left|\sum_{t=0}^{T-1} u^\top \frac{\partial^2}{\partial^2\theta}\pi_\theta(A_t|S_t)u\right|$$

$$\leq 5TB^2\|u\|_2^2. \qquad \text{(By (22))}$$

As for (b) in (20),

$$|(b)| = \left|\sum_{t=0}^{T-1}\left\langle\frac{\partial}{\partial\theta}\pi_\theta(A_t|S_t), u\right\rangle \cdot \sum_{t\neq t'}\left\langle\frac{\partial}{\partial\theta}\pi_\theta(A_{t'}|S_{t'}), u\right\rangle \prod_{t''\neq t,t'}\pi_\theta(A_{t''}|S_{t''})\right|$$

$$\leq \sum_{t=0}^{T-1}\left|\left\langle\frac{\partial}{\partial\theta}\pi_\theta(A_t|S_t), u\right\rangle\right| \cdot \sum_{t\neq t'}\left|\left\langle\frac{\partial}{\partial\theta}\pi_\theta(A_{t'}|S_{t'}), u\right\rangle\right| \cdot 1$$

$$\leq 2TB\|u\|_2 \cdot 2TB\|u\|_F \qquad \text{(By (21))}$$

$$= 4T^2B^2\|u\|_2^2.$$

Thus, looking back at the (20), we have

$$\left|u^\top \frac{\partial^2}{\partial^2\theta}w_{\pi_\theta}(h)u\right|$$

$$\leq |(a)| + |(b)|$$

$$\leq 5TB^2\|u\|_2^2 + 4T^2B^2\|u\|_2^2.$$

Therefore,

$$\left| \text{IS}(\pi_e, \pi_\theta, H)^2 u^\top \frac{\partial^2}{\partial^2 \theta} w_{\pi_\theta}(h) u \right|$$

$$\leq \left| \text{IS}(\pi_e, \pi_\theta, H)^2 \right| \cdot \left| u^\top \frac{\partial^2}{\partial^2 \theta} w_{\pi_\theta}(h) u \right|$$

$$\leq C^{2T} T^3 B^2 \|u\|_2^2 (5F + 4T). \tag{23}$$

Putting these all together,

$$\left| \frac{\partial^2}{\partial^2 \alpha} \mathbb{V}_{H \sim p_\omega, \pi_{\theta_\alpha}}[\text{IS}(\pi_e, \pi_{\theta_\alpha}, H)] \Big|_{\alpha=0} \right|$$

$$= \left| \sum_h -\tilde{p}(h) \left[ \left\langle \frac{\partial}{\partial \theta} \text{IS}(\pi_e, \pi_\theta, H)^2, u \right\rangle \left\langle \frac{\partial}{\partial \theta} w_{\pi_\theta}(h), u \right\rangle + \text{IS}(\pi_e, \pi_\theta, H)^2 u^\top \frac{\partial^2}{\partial^2 \theta} w_{\pi_\theta}(h) u \right] \right|$$
$$\text{(By (17))}$$

$$\leq \left| \left\langle \frac{\partial}{\partial \theta} \text{IS}(\pi_e, \pi_\theta, H)^2, u \right\rangle \left\langle \frac{\partial}{\partial \theta} w_{\pi_\theta}(h), u \right\rangle \right| + \left| \text{IS}(\pi_e, \pi_\theta, H)^2 u^\top \frac{\partial^2}{\partial^2 \theta} w_{\pi_\theta}(h) u \right|$$

$$\leq \left( 2\sqrt{2} B T^3 C^{2T} \|u\|_2 \right) \cdot \left( \sqrt{2} B T \|u\|_2 \right) + C^{2T} T^3 B^2 \|u\|_2^2 (5F + 4T) \quad \text{(By (18)(19)(23))}$$

$$= 4 B^2 C^{2T} T^4 \|u\|_2^2 + C^{2T} T^3 B^2 \|u\|_2^2 \cdot (5 + 4T)$$

$$= B^2 C^{2T} T^3 \|u\|_2^2 (5 + 8T).$$

Thus,

$$\left\| \frac{\partial^2}{\partial^2 \theta} \mathbb{V}_{H \sim p_\omega, \pi_\theta}[\text{IS}(\pi_e, \pi_\theta, H)] \right\|_{\text{op}} = \sup_{\|u\|_2 = 1} B^2 C^{2T} T^3 \|u\|_2^2 (5 + 8T)$$

$$= B^2 C^{2T} T^3 (5 + 8T),$$

where $\|\cdot\|_{\text{op}}$ denotes the operator norm. Therefore, we conclude that the objective function $\mathbb{V}_{H \sim p_\omega, \pi_\theta}[\text{IS}(\pi_e, \pi_\theta, H)]$ is $\ell_\Theta$-smooth in $\theta$ with $\ell_\Theta = B^2 C^{2T} T^3 (5 + 8Td)$.
Lastly, the convexity of the objective function follows directly from Lemma 2 of Hanna et al. (2024). □

### A.8 PROOF OF LEMMA 4

*Proof.* We first show that $\Phi(\theta)$ is $L_\Theta$-Lipschitz in $\theta$. By Lemma 3, we know that $\mathbb{V}_{H \sim p_\omega, \pi_\theta}[\text{IS}(\pi_e, \pi_\theta, H)]$ is $L_\Theta$-Lipschitz.
$\forall \theta_1, \theta_2 \in \Theta$, define $p_{\omega_1} \doteq \arg\max_{p_\omega} \mathbb{V}_{H \sim p_\omega, \pi_{\theta_1}}[\text{IS}(\pi_e, \pi_{\theta_1}, H)]$, $p_{\omega_2} \doteq \arg\max_{p_\omega} \mathbb{V}_{H \sim p_\omega, \pi_{\theta_2}}[\text{IS}(\pi_e, \pi_{\theta_2}, H)]$. Then,

$$\Phi(\theta_1) - \Phi(\theta_2) = \max_{p_\omega} \mathbb{V}_{H \sim p_\omega, \pi_{\theta_1}}[\text{IS}(\pi_e, \pi_{\theta_1}, H)] - \max_{p_\omega} \mathbb{V}_{H \sim p_\omega, \pi_{\theta_2}}[\text{IS}(\pi_e, \pi_{\theta_2}, H)]$$

$$= \mathbb{V}_{H \sim p_{\omega_1}, \pi_{\theta_1}}[\text{IS}(\pi_e, \pi_{\theta_1}, H)] - \mathbb{V}_{H \sim p_{\omega_2}, \pi_{\theta_2}}[\text{IS}(\pi_e, \pi_{\theta_2}, H)]$$

$$\leq \mathbb{V}_{H \sim p_{\omega_1}, \pi_{\theta_1}}[\text{IS}(\pi_e, \pi_{\theta_1}, H)] - \mathbb{V}_{H \sim p_{\omega_1}, \pi_{\theta_2}}[\text{IS}(\pi_e, \pi_{\theta_2}, H)]$$

$$\leq L_\Theta \|\theta_1 - \theta_2\|. \quad \text{(By Lemma 3)}$$

By symmetry, with also have

$$\Phi(\theta_2) - \Phi(\theta_1) \leq L_\Theta \|\theta_1 - \theta_2\|.$$

Thus,

$$|\Phi(\theta_1) - \Phi(\theta_2)| \leq L_\Theta \|\theta_1 - \theta_2\|,$$

which shows the Lipschitz property.
Next, from Lemma 3, we also know that $\mathbb{V}_{H \sim p_\omega, \pi_\theta}[\text{IS}(\pi_e, \pi_\theta, H)]$ is convex in $\theta$ under the linear

softmax parameterization of the behavior policy $\pi_\theta$. Thus, $\forall \theta_1, \theta_2 \in \Theta$ and $t \in [0, 1]$,

$$
\begin{aligned}
\Phi(t\theta_1 + (1-t)\theta_2) &= \max_{p_\omega} \mathbb{V}_{H \sim p_\omega, \pi_{(t\theta_1 + (1-t)\theta_2)}}[\mathrm{IS}(\pi_e, \pi_{(t\theta_1 + (1-t)\theta_2)}, H)] \\
&\leq \max_{p_\omega}[t\mathbb{V}_{H \sim p_\omega, \pi_{\theta_1}}[\mathrm{IS}(\pi_e, \pi_{\theta_1}, H)] + (1-t)\mathbb{V}_{H \sim p_\omega, \pi_{\theta_2}}[\mathrm{IS}(\pi_e, \pi_{\theta_2}, H)]] \\
&\qquad\qquad\qquad\qquad\qquad\qquad\qquad\qquad\qquad\qquad\qquad\qquad \text{(By Lemma 3)} \\
&\leq t\max_{p_\omega}[\mathbb{V}_{H \sim p_\omega, \pi_{\theta_1}}[\mathrm{IS}(\pi_e, \pi_{\theta_1}, H)] + (1-t)\max_{p_{\omega'}} \mathbb{V}_{H \sim p_{\omega'}, \pi_{\theta_2}}[\mathrm{IS}(\pi_e, \pi_{\theta_2}, H)] \\
&= t\Phi(\theta_1) + (1-t)\Phi(\theta_2).
\end{aligned}
$$

Therefore, we show that $\Phi(\theta)$ is convex in $\theta$. $\qquad\square$

## A.9 PROOF OF THEOREM 5

*Proof.* To begin with, we define $\theta^* \doteq \arg\min_{\theta \in \Theta} \Phi(\theta)$. Since the set $\Theta$ is closed and convex, the Euclidean projection is nonexpensive. That is, $\forall u \in \mathbb{R}^d, z \in \Theta$,

$$
\|\mathrm{Proj}_\Theta(u) - z\|^2 \leq \|u - z\|^2.
$$

With $u \doteq \theta_i - \alpha\mathcal{G}_i$, $z \doteq \theta^*$, we have

$$
\mathrm{Proj}_\Theta(u) = \theta_{i+1}.
$$

Thus,

$$
\begin{aligned}
\|\theta_{i+1} - \theta^*\|^2 &\leq \|\theta_i - \alpha\mathcal{G}_i - \theta^*\|^2 \\
&= \|\theta_i - \theta^*\|^2 - 2\alpha\langle\mathcal{G}_i, \theta_i - \theta^*\rangle + \alpha^2\|\mathcal{G}_i\|^2. \qquad (24)
\end{aligned}
$$

From here, we first bound the last term, $\|\mathcal{G}_i\|^2$. By Lemma 3 we know that the objective function $\mathbb{V}_{H \sim p_\omega, \pi_\theta}[\mathrm{IS}(\pi_e, \pi_\theta, H)]$ is $L_\Theta$-Lipschitz and convex in $\theta$. Thus we have

$$
\|\mathcal{G}_i\| \leq L_\Theta \implies \|\mathcal{G}_i\|^2 \leq L_\Theta^2. \qquad (25)
$$

Next, since the gradient objective $\mathbb{V}_{H \sim p_\omega, \pi_\theta}[\mathrm{IS}(\pi_e, \pi_\theta, H)]$ is differentiable and convex in $\theta$, we have the subgradient inequality that

$$
\left\langle \mathcal{G}_i, \theta_i - \theta^* \right\rangle \geq \mathbb{V}_{H \sim p_{\omega_i}, \pi_{\theta_i}}[\mathrm{IS}(\pi_e, \pi_{\theta_i}, H)] - \mathbb{V}_{H \sim p_{\omega_i}, \pi_{\theta^*}}[\mathrm{IS}(\pi_e, \pi_{\theta^*}, H)].
$$

Remember that we defined

$$
\Phi(\theta) \doteq \max_{p_\omega} \mathbb{V}_{H \sim p_\omega, \pi_\theta}[\mathrm{IS}(\pi_e, \pi_\theta, H)].
$$

Thus,

$$
\mathbb{V}_{H \sim p_{\omega_i}, \pi_{\theta^*}}[\mathrm{IS}(\pi_e, \pi_{\theta^*}, H)] \leq \Phi(\theta^*),
$$

and by Algorithm 2,

$$
\max_p \mathbb{V}_{H \sim p, \pi_{\theta_i}}[\mathrm{IS}(\pi_e, \pi_{\theta_i}, p, H)] = \Phi(\theta_i) \leq \mathbb{V}_{H \sim p_{\omega_i}, \pi_{\theta_i}}[\mathrm{IS}(\pi_e, \pi_{\theta_i}, H)] + \epsilon_i.
$$

Therefore,

$$
\left\langle \mathcal{G}_i, \theta_i - \theta^* \right\rangle \geq \Phi(\theta_i) - \epsilon_i - \Phi(\theta^*). \qquad (26)
$$

Putting it all together, by (24), (25), and (26), we have

$$
\|\theta_{i+1} - \theta^*\|^2 \leq \|\theta_i - \theta^*\|^2 - 2\alpha(\Phi(\theta_i) - \epsilon_i - \Phi(\theta^*)) + \alpha^2 L_\Theta^2.
$$

Rearranging the terms, we get

$$
2\alpha[\Phi(\theta_i) - \Phi(\theta^*)] \leq \|\theta_i - \theta^*\|^2 - \|\theta_{i+1} - \theta^*\|^2 + 2\alpha\epsilon_i + \alpha^2 L_\Theta^2.
$$

Taking the summation over $i$,

$$2\alpha \sum_{i=0}^{n-1} \Phi(\theta_i) - \Phi(\theta^*) \leq \|\theta_0 - \theta^*\|^2 + 2\alpha \sum_{i=0}^{n-1} \epsilon_i + n\alpha^2 L_\Theta^2.$$

Since $\theta_0, \theta^* \in \Theta$, and we defined $\mathrm{diam}(\Theta) \leq D$ where

$$\mathrm{diam}(\Theta) \doteq \max_{\theta, \theta' \in \Theta} \|\theta - \theta'\|,$$

we have

$$\|\theta_0 - \theta^*\|^2 \leq D^2.$$

Thus,

$$\frac{1}{n} \sum_{i=0}^{n-1} \Phi(\theta_i) - \Phi(\theta^*) \leq \frac{D^2}{2\alpha n} + \frac{\alpha L_\Theta^2}{2} + \frac{1}{n} \sum_{i=0}^{n-1} \epsilon_i. \tag{27}$$

According to Algorithm 2,

$$\bar{\theta} \doteq \frac{1}{n} \sum_{i=0}^{n-1} \theta_i.$$

By Lemma 4, $\Phi(\pi_\theta)$ is convex in $\theta$. Thus, by induction with the basic convex property, with the nonnegative weight $\frac{1}{n}$ and the fact that $\sum_{i=0}^{n-1} \frac{1}{n} = 1$, we obtain

$$\Phi(\bar{\theta}) = \Phi\left(\frac{1}{n} \sum_{i=0}^{n-1} \theta_i\right) \leq \frac{1}{n} \sum_{i=0}^{n-1} \Phi(\theta_i).$$

Subtracting $\Phi(\theta^*)$ form both sides, we get

$$\Phi(\bar{\theta}) - \Phi(\theta^*) \leq \frac{1}{n} \sum_{i=0}^{n-1} \Phi(\theta_i) - \Phi(\theta^*).$$

Plugging it into (27),

$$\Phi(\bar{\theta}) - \Phi(\theta^*) \leq \frac{D^2}{2\alpha n} + \frac{\alpha L_\Theta^2}{2} + \frac{1}{n} \sum_{i=0}^{n-1} \epsilon_i.$$

With the definition $\theta^* \doteq \arg\min_{\theta \in \Theta} \Phi(\theta)$ and $\alpha \doteq \frac{D}{L_\Theta \sqrt{n}}$, we then have

$$\Phi(\bar{\theta}) - \min_{\theta \in \Theta} \Phi(\theta) \leq \frac{D L_\Theta}{\sqrt{n}} + \frac{1}{n} \sum_{i=0}^{n-1} \epsilon_i.$$

$\square$

# B NUMERICAL STUDIES

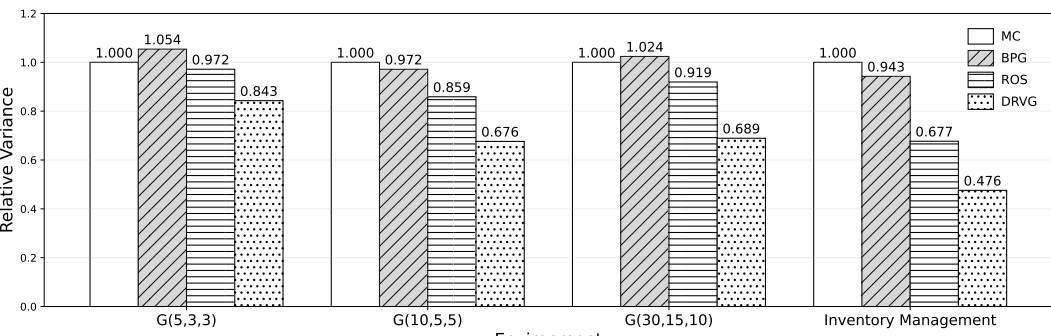

Figure 3: Supplementary figure for Section 6.1. Relative variance of each method under its tailored adversarial transition. All values are normalized by the variance of the on-policy Monte Carlo (MC) method (under its tailored adversarial transition) in the same environment.

In our numerical studies, we leverage a wide range of target policies ranging from completely random to highly deterministic. Specifically, a policy $\pi_{\text{train}}$ is computed as the optimal policy of the MDP model, and $\pi_{\text{random}}$ is randomly generated. Then, the target policies $\pi_e$ are set to be $(1 - \beta)\pi_{\text{train}} + \beta\pi_{\text{random}}$ with $\beta \in \{\frac{1}{30}, \frac{2}{30}, ..., 1\}$. For each of the 30 target policies, we have 30 independent runs, resulting in a total of 900 runs for each value.

In Section 6.1, we generate method-specific adversarial transitions by running Algorithm 1 separately for each method, yielding one adversarial transition per behavior policy. For ROS (Zhong et al., 2022), which adapts its behavior policy, we generate the adversarial transition by treating the target policy as the nominal behavior policy in Algorithm 1. We first measure each method's variance under the original simulator transition $p_0$, and then under its method-specific adversarial transition $p_{\text{adv}}$. Note that $p_{\text{adv}}$ differs across methods. We report the relative variance of each method under its own $p_{\text{adv}}$ in Figure 3, where each value is normalized by the variance of MC (also under its $p_{\text{adv}}$) for the same environment. Finally, we report the variance increase for each method in Figure 1, defined as the difference between its variance under $p_{\text{adv}}$ and under $p_0$. In Section 6.2, we evaluate all methods under a shared adversarial target transition. This transition is constructed by applying Algorithm 1 to the on-policy Monte Carlo baseline. We then report each method's variance under this transition in Figure 2.

## B.1 GARNET EXAMPLES

A Garnet environment (Archibald et al., 1995) is represented by three integers $(|S|, |A|, b)$, denoting the number of states, actions, and the branching factor, respectively. By varying $b$, one controls the degree of stochasticity: small $b$ yields sparse transitions, while large $b$ approaches fully connected transitions. This flexibility makes Garnets particularly suitable for stress-testing reinforcement learning algorithms across a wide spectrum of transition structures (Tarbouriech and Lazaric, 2019; Wang et al., 2023a;b). We evaluate the four methods on three Garnet instances—$G(5, 3, 3)$, $G(10, 5, 5)$, and $G(30, 15, 10)$—which span increasing environment sizes and connectivity levels.

## B.2 INVENTORY MANAGEMENT

Inventory management (Porteus, 2002; Ho et al., 2018) is a classical stochastic control problem under transition uncertainty. The state corresponds to inventory levels, actions represent order quantities, and stochastic demand drives the state transitions. In our inventory management example, we adopt radial-type basis functions as introduced in Sutton and Barto (2018), defined for state $s$ and feature index $i$ as $\phi_i(s) = \exp\left(-\frac{\|s - c_i\|^2}{2\sigma_i^2}\right)$, where $c_i$ and $\sigma_i$ denote the deterministic center and scaling parameter of the $i$-th feature, respectively. This nonlinear parameterization captures variations in state representation while controlling the expressive capacity of the model under uncertainty.

