# OpenReview forum: "Efficient and Robust Behavior Policy Search for Online Off-policy Evaluation through Transition Gradients"
_ICLR.cc/2026/Conference — Submitted to ICLR 2026_

### Official Review · Reviewer_qc4C · 2025-10-18

**Soundness:** 3
**Presentation:** 3
**Contribution:** 3
**Rating:** 6
**Confidence:** 3

**Summary:**

This paper introduces a robust and efficient behavior policy search (BPS) framework for reinforcement learning (RL) policy evaluation under transition uncertainty. Traditional on-policy Monte Carlo evaluation suffers from high variance, and standard BPS methods typically assume fixed simulator dynamics, which can lead to poor robustness for real-world applications.

This paper proposes a novel minimax formulation, where the inner-layer (i.e. maximization) represents adversarial transition perturbations and the outer-layer (i.e. minimization) solves for a behavior policy that minimizes worst-case evaluation variance.
The paper derives closed-form transition-gradient expressions (on-transition and off-transition cases), designs an inner-loop adversarial transition optimizer, and proposes a double-loop robust gradient algorithm (DRVG) for variance-minimizing policy search. Theoretically, they prove global convergence under linear-softmax parameterization with an $O(1/\sqrt{n})$rate.

The paper then conducts experiments on Garnet MDPs and an inventory management task, which show that DRVG has lower variance and is more robust to adversarial transition perturbations than baselines including MC, Behavior Policy Gradient, and Robust On-policy Sampling.

**Strengths:**

**Originality**: good

1, The paper discusses behavior policy search from a minimax optimization point of view with transition uncertainty, which seems novel in the RL evaluation literature to me.

2, It introduces transition-gradient methods for variance objectives. This seems a direction not thorough explored previously.

3, The unification of variance reduction and robustness bridges off-policy evaluation and robust MDP theory, a meaningful conceptual contribution.

**Quality**: fine

1, Theoretically rigorous paper: multiple analytical gradient theorems are presented with proofs and convergence guarantees.

2, Empirical results, though limited in scale, consistently support the theoretical claims.

**Clarity**: good

1, The math of the paper is clear; definitions are presented without ambiguity,

2, key steps are summarized in theorem boxes and algorithms.

**Significance**: fine

1, Robust policy evaluation is crucial for sim-to-real transfer and safe RL; the proposed method contributes toward that.

2, The convergence guarantee for a variance-minimizing, nonconvex–nonconcave min-max problem is an important theoretical advancement.

**Weaknesses:**

There is no critical weakness detected. However, I do have the following concerns.

(1) Some assumptions seem a bit strong.
Coverage/bounded-ratio assumptions are strong and under-discussed. Convergence and smoothness rely on a uniform bound $C$ on quantities such as importance ratios, plus bounded features, etc. Otherwise the current analysis doesn’t apply. I am not sure if these assumptions are realistic.

(2) Lack of ablation / sensitivity studies

The role of the KL regularization coefficient $\eta$ is not empirically analyzed; varying $\eta$ could show the trade-off between robustness and over-conservatism.

The effect of inner-loop precision $\epsilon_i$ on performance and convergence is not reported.

(3) (minor weakness), Limited empirical experiment

Experiments are restricted to synthetic MDPs (Garnet) and a toy control problem (inventory management).

**Questions:**

(1) The global convergence theorem (Theorem 5) assumes a linear–softmax form and convex domain. Do the authors believe the same convergence behavior holds in the nonconvex case (either theoretically or empirically)? Would like to hear some comments.

(2) The proposed algorithm claims to achieve ‘robustness’. What is exactly the definition of ‘robustness’? Is it equivalent to ‘variance minimization’?

(3) The entire paper focuses on variance minimization and assumes unbiasedness of the OPE estimator. I might miss something but why can we take unbiasedness for granted?


Thank you.

---

> ### Author Response · Authors · 2025-11-22
>
> Many thanks for the encouraging and thoughtful feedback! We are grateful that the reviewer highlights the novelty of our minimax formulation, the contribution of transition-gradient methods, and the meaningful unification of robustness and variance reduction. We also appreciate the recognition of the rigor and clarity of our theoretical development, as well as the empirical results that support our claims. We answer each remaining question in detail below.
>
> For Weaknesses:
>
> 1. >Some assumptions seem a bit strong.
>
> Thank you for raising this point. The assumptions you mentioned—coverage, bounded importance-sampling ratios, and bounded features—are in fact standard and **widely used** in the policy-gradient and off-policy evaluation literature. In particular, bounded IS ratios are commonly assumed in behavior-policy search and variance analyses (e.g., [1][2]) to ensure the variance objective is well defined. Practically, this assumption can always be satisfied by requiring the behavior policy $\pi_\theta$​ to be bounded away from zero on the support of the target policy.
>
> The bounded-feature condition is also common in convergence proofs for policy-gradient methods and robust MDP analyses (e.g., [3][4]), where they enable Lipschitz and smoothness arguments needed for gradient-based optimization. These assumptions are generally mild and reflect standard modeling choices.
>
> 2. >The role of the KL regularization coefficient is not analyzed … The effect of inner-loop precision  on performance and convergence is not reported.
>
> Thank you for this helpful suggestion. As you pointed out, in our formulation, the KL coefficient $\eta$ controls the strength of the KL regularization in the inner loop and therefore governs the trade-off between robustness and fidelity to the simulator. Specifically, a smaller $\eta$ permits larger adversarial deviations (yielding stronger robustness), while a larger $\eta$ keeps the adversary closer to the original simulator's transition $p_{\omega_0}$​​. This robustness–fidelity balance is standard in KL-regularized RL; for instance, PPO adjusts its KL penalty coefficient to target a desired trust-region size [5], and Vieillard et al. [6] analyze how KL weights influence stability and conservatism. In practice, the appropriate $\eta$ depends on how much the practitioner trusts the simulator, and there is **no problem-agnostic “best” setting**.
>
> Regarding the inner-loop precision $\epsilon$, this parameter controls how accurately the adversarial transition is optimized before updating the behavior policy. Such tolerance parameters are standard in nested optimization methods (e.g., [7]), where the inner maximization only needs to be solved to a sufficient degree so that its ascent direction reliably guides the outer update. Because our inner problem is smooth and regularized by the KL term, the maximizer is stable, and the outer loop remains well behaved over a wide range of $\epsilon$. In practice, RL practitioners may simply run a fixed small number of inner ascent steps per outer update, which provides a sufficiently accurate adversarial direction while avoiding unnecessary computational overhead.
>
> 3. >Experiments are restricted to synthetic MDPs (Garnet) and a toy control problem (inventory management).
>
> Thanks for the constructive comment. Our paper develops a unified framework for efficient and robust policy evaluation. It provides a minimax formulation, analytical transition-variance gradients, and a global convergence guarantee for behavior-policy search under adversarial transitions. The experiments are designed as proof-of-concept demonstrations, showing that the method consistently improves robustness when transitions are perturbed.
>
> We also note that using small or medium-scale environments is standard practice in robust-RL papers that introduce new theoretical frameworks. Recent works in the robust-RL community ([7][12]) evaluate on Garnet MDPs that are even smaller than those used in our paper, and some omit the inventory-management setting entirely. The comparison is summarized in the table below:
>
> |               | Garnet | Inventory Management |
> |---------------|---------------|-------------|
> | **Ours**      |  $G(30,15)$      | Yes |
> | [7] (ICML 2023) | $G(15,8) $  |  Yes |
> | [12] (Neurips 2024)    | $G(10,5)$       | No |
>
> *Table 1:  Environments used in recent robust-RL works. Larger $G(\cdot,\cdot)$ settings reflect more challenging Garnet tasks, and our experiments operate at comparable or higher complexity.*

---

> ### Author Response · Authors · 2025-11-22
>
> For Questions:
>
> 1. >Do the authors believe the same convergence behavior holds in the nonconvex case (either theoretically or empirically)?
>
> Thank you for this insightful question! As widely noted in the optimization and robust-RL literature, min–max objectives of the form we study are generally nonconvex–nonconcave, and global convergence guarantees are typically unattainable without additional structure (e.g., [8][9][10]). For this reason, it is standard practice in theoretical analyses to adopt structured parameterizations, such as linear or linear–softmax policies, that render the objective well behaved and allow global convergence results to be established (e.g.,[1][2][3][7]). Our use of the linear–softmax model **follows this convention** and enables a clean and rigorous analysis.
>
> Regarding the case without the linear–softmax assumption, global convergence guarantees for fully nonconvex–nonconcave minimax optimization remain a challenging open problem, as also noted in prior work [8][9][10]. Nonetheless, in our experimental section we implement the algorithm with a **more expressive neural-network parameterization** and observe improved robustness compared with non-robust baselines. This suggests that the desirable robust behavior extends beyond the structured setting used in the theoretical analysis.
>
> 2. >The proposed algorithm claims to achieve ‘robustness’. What is exactly the definition of ‘robustness’? Is it equivalent to ‘variance minimization’?
>
> Thanks for this thoughtful question. In our paper, “robustness” is not defined as variance minimization alone. Our goal is to obtain efficient policy evaluation under potentially perturbed transitions, and this is captured by the minimax formulation in Eqs. (1)–(2). The inner maximization models worst-case transition perturbations, reflecting potential simulator–environment mismatch, while the outer minimization finds a behavior policy whose evaluation **variance remains low** even under such adversarial transitions.
>
> Thus, robustness in our work refers to maintaining low variance under adversarial transition shifts. The minimax formulation therefore unifies efficiency (through variance minimization) and robustness (through adversarial variance maximization), which is the central goal of our approach.
>
> 3. >The entire paper focuses on variance minimization and assumes unbiasedness of the OPE estimator. I might miss something but why can we take unbiasedness for granted?
>
> Many thanks for your comment! Our use of unbiased OPE estimators follows standard practice in the behavior-policy search (BPS) literature, and the role of unbiasedness differs depending on whether the transition is **fixed or perturbed**. Specifically,
>
> **(1)** Fixed Transition
>
> We adopt the same modeling choice as prior BPS works [1][2], which assume unbiased OPE estimators and study behavior-policy search through variance minimization. In fact, many commonly used estimators, such as the importance sampling (IS), are unbiased whenever the behavior policy provides sufficient coverage, as is standard in OPE theory (e.g., Theorem 1 in [11]).
>
> **(2)** Perturbed Transition
>
> Once transitions are perturbed due to mismatches between the simulator and the real world, the true value of the target policy also changes with the transition shift. Thus, the notion of unbiasedness does not apply in the robust part of our formulation.
> We appreciate the reviewer raising this point. Extending our efficient-and-robust evaluation framework to biased estimators—such as bootstrapping estimators—would indeed be an interesting direction for future work, and we thank the reviewer for pointing out this opportunity!
>
> We hope our responses address all remaining concerns, and we would be very happy to engage in further discussion if that would be helpful.

---

> ### Author Response · Authors · 2025-11-22
>
> **References**:
>
> [1] Josiah P. Hanna, Yash Chandak, Philip S. Thomas, Martha White, Peter Stone, Scott Niekum.
> "Data-Efficient Policy Evaluation Through Behavior Policy Search" (ICML 2017)
>
> [2] Josiah P. Hanna, Yash Chandak, Philip S. Thomas, Martha White, Peter Stone, Scott Niekum.
> "Data-Efficient Policy Evaluation Through Behavior Policy Search" (JMLR 2024)
>
> [3] Alekh Agarwal, Sham Kakade, Jason Lee, Gaurav Mahajan.
> "On the Theory of Policy Gradient Methods: Optimality, Approximation, and Distribution Shift" (JMLR 2021)
>
> [4] Matthew S. Zhang, Murat A. Erdogdu, Animesh Garg.
> Convergence and Optimality of Policy Gradient Methods in Weakly Smooth Settings. (AAAI 2022)
>
> [5] John Schulman, Filip Wolski, Prafulla Dhariwal, Alec Radford, Oleg Klimov.
> "Proximal Policy Optimization Algorithms" (PPO)
>
> [6] Nino Vieillard, Tadashi Kozuno, Bruno Scherrer, Olivier Pietquin, Rémi Munos, Matthieu Geist.
> "Leverage the Average: an Analysis of KL Regularization in Reinforcement Learning"  (NeurIPS 2020)
>
> [7] Qiuhao Wang, Chin Pang Ho, Marek Petrik.
>      "Policy Gradient in Robust MDPs with Global Convergence Guarantee" (ICML 2023)
>
> [8] Moustafa Nouiehed, Meisam Razaviyayn.
> “Solving a Class of Non-Convex Min-Max Games Using Iterative First-Order Methods” (NeurIPS 2019)
>
> [9] Chi Jin, Praneeth Netrapalli, Michael I. Jordan.
> “What Is Local Optimality in Nonconvex–Nonconcave Minimax Optimization?” (ICML 2020)
>
> [10] Yifei Zheng, Samet Oymak, Yin Tat Lee.
> “Universal Gradient Descent–Ascent for Nonconvex–Nonconcave Minimax Problems” (NeurIPS 2023)
>
> [11] Shuze Liu, Shangtong Zhang. “Improving Monte Carlo Evaluation with Offline Data” (ICML 2024)
>
> [12] Zhongchang Sun, Sihong He, Fei Miao, Shaofeng Zou.
> "Policy Optimization for Robust Average Cost MDPs" (NeurIPS 2024)

---

> ### Comment · Reviewer_qc4C · 2025-11-26
> **Re: Rebuttal**
>
> I thank the authors for responding to my concerns about the paper.
>
> Specifically,
> (1) regarding the weakness of strong mathematical assumption: I can buy the argument that these are standard assumptions.
>
> (2) I appreciate the clarification regarding the KL coefficient.
>
> (3) The limitation to small synthetic environment is not ideal but fine.
>
> Overall, I do not have further concerns about the paper. I will maintain my score of 6 and recommend acceptance.
>
> Thank you!

---

> > ### Author Response · Authors · 2025-11-26
> >
> > Thank you very much for your thoughtful review and for your positive recommendation. We’re glad that our clarifications addressed your concerns, and we truly appreciate your detailed feedback on the assumptions, KL coefficient, and experimental setup! We belive it will help us further for future work.

---

### Official Review · Reviewer_eKd7 · 2025-10-28

**Soundness:** 3
**Presentation:** 2
**Contribution:** 2
**Rating:** 4
**Confidence:** 3

**Summary:**

The paper studies behavior policy search (BPS) for off-policy evaluation (OPE) under transition-model misspecification. It frames BPS as a min--max objective that seeks behavior policies whose evaluation variance is small even under adversarially perturbed dynamics within a KL-bounded ambiguity set. The authors derive transition-variance gradient formulas for both on-transition and off-transition forms, propose an inner-loop adversarial ascent over transition parameters and an outer-loop projected descent for the behavior policy and provide a global convergence guarantee under linear-softmax policies with bounded features. Experiments on Garnet MDPs and an inventory-management task suggest improved robustness (lower variance, less sensitivity to transition shifts) compared to baselines such as Monte Carlo, Behavior Policy Gradient (BPG), and ROS.

**Strengths:**

- This paper develops a logically consistent double-loop optimization framework that integrates behavior policy search under transition perturbations, analytical gradient derivation, and global convergence analysis.

- Unlike conventional BPS methods assuming fixed transitions, the authors introduce an adversarial perturbation model to capture simulator–environment discrepancies, theoretically enhancing robustness to transition uncertainty and unifying OPE and RMDP principles.

- The derived gradient expressions are estimator-independent and combined with a KL regularizer to constrain adversarial deviations, ensuring both robustness and extensibility to advanced estimators or real-world deployment.

**Weaknesses:**

- A key theoretical inconsistency arises in the formulation of the proposed min--max objective, defined in Eq. (1)--(2) as $\min_{\theta \in \Theta} \max_{\omega \in \Omega} V_{H \sim p_{\omega}, \pi_{\theta}} [\mathrm{OPE}(\pi_e, \pi_{\theta}, H)].$ This objective is inherently nonconvex--nonconcave and generally non-differentiable with respect to both the policy parameter $\theta$ and the transition parameter $\omega$. The authors acknowledge this challenge (lines345--350), yet later claim a global convergence guarantee in Theorem 5. However, this guarantee is established only under a restrictive linear-softmax policy parameterization, which artificially convexifies the objective and simplifies the underlying problem.
- The theoretical results hinge on the linear-softmax policy assumption (lines358--361). While this setting is analytically convenient, it is limited to linear function approximation, a regime that has been extensively explored in earlier literature. In contrast, the experimental section employs neural network parameterizations (line452), which invalidate the convexity assumptions required for the claimed convergence. As a result, the global convergence guarantee does not strictly apply to the practical algorithm used in the experiments.
- Although the paper emphasizes theoretical robustness, the empirical evidence remains limited. The experiments focus mainly on small-scale settings such as Garnet MDP and inventory management. To convincingly demonstrate the robustness and scalability of the proposed double-loop DRVG algorithm under transition uncertainty, evaluations on more complex and high-dimensional environments (e.g., MuJoCo, Atari, or robotic simulators like Gazebo) would be valuable. Such experiments would better substantiate the claimed robustness of the minimax formulation.

**Questions:**

- Why does the paper adopt a standard MDP formulation instead of a more general POMDP framework, given that transition uncertainty or partial observability could naturally arise in the studied setting?

- For better readability and referencing (e.g., in Section 5.2), it would be beneficial to formally present key conditions using a numbered `assumption` environment.

- The paper introduces the regularization parameter $\eta$ in the KL-penalized inner-loop objective, yet does not explain how $\eta$ is selected in practice. Could the authors provide intuition or a principled guideline for tuning $\eta$? For example, how sensitive is performance to this choice, and does it correspond to a trade-off between robustness and fidelity to the simulator dynamics?

---

> ### Author Response · Authors · 2025-11-22
>
> Thank you for your detailed review and practical suggestions. Your comments show that our work is well written, and that the robustness of our method is supported both theoretically and empirically. We appreciate the reviewer’s thoughtful assessment of our formulation and analysis, and we respond to each concern below.
>
> For Weaknesses
>
> 1&2.
> >A key theoretical inconsistency arises in the formulation of the proposed min--max objective.
>
> >As a result, the global convergence guarantee does not strictly apply to the practical algorithm used in the experiments.
>
> We appreciate the reviewer’s detailed reading and would like to clarify that our formulation is not a theoretical inconsistency. Min–max objectives in robust RL and adversarial MDPs are typically nonconvex–nonconcave unless special structural assumptions are imposed, a difficulty widely acknowledged in the optimization and robust-RL literature [1][2][3]. Because global guarantees are known to be unattainable without such structure, it is **standard practice** to analyze convergence under structural assumptions, such as linear or linear-softmax parameterization, that render the problem well-structured (e.g., see [4](JMLR 2021), [5](ICML 2023), and [6](JMLR 2024)). Our use of the linear-softmax transitions **follows this convention** and enables a clean theoretical analysis.
>
> Importantly, our empirical evaluation uses more expressive neural-network parameterizations, demonstrating that the algorithm remains effective even beyond the structured theoretical setting. This separation between a tractable model for theory and a more general architecture for practice is common and **widely accepted** in the RL theory literature (e.g., see [7](NeurIPS 2020) and [8](NeurIPS 2019)). Thus, we believe this does not represent a theoretical inconsistency, but aligns with the standard modeling practice adopted in many influential RL works.
>
> 3. >Although the paper emphasizes theoretical robustness, the empirical evidence remains limited.
>
> Thank you for the constructive comment! Our paper introduces a unified framework for efficient and robust policy evaluation, combining a minimax formulation, analytical transition-variance gradients, and the first global convergence result for behavior-policy search under adversarial dynamics. The experiments serve as proof-of-concept demonstrations, showing that the method consistently improves robustness under transition perturbations.
>
> Besides, we would like to note that the use of small to medium-scale environments is in fact standard in papers that establish new theoretical frameworks in robust RL. For instance, some **recent robust-RL papers** such as [5][12] use Garnet environments, but with even smaller sizes than the one used by our paper. Also, some of them do not incorporate the classical stochastic-transition inventory management example. For completeness, we restate the comparison (Table 1) below:
>
>
> |               | Garnet | Inventory Management |
> |---------------|---------------|-------------|
> | **Ours**      |  $G(30,15)$      | Yes |
> | [5] (ICML 2023) | $G(15,8) $  |  Yes |
> | [12] (Neurips 2024)    | $G(10,5)$       | No |
>
> *Table 1:  Environments used in recent robust-RL works. Larger $G(\cdot,\cdot)$ settings reflect more challenging Garnet tasks, and our experiments operate at comparable or higher complexity.*
>
>
> **For Questions**
>
> 1. >Why does the paper adopt a standard MDP formulation instead of a more general POMDP framework?
>
> We thank the reviewer for the insightful question. We adopt the standard MDP formulation because fully observable MDPs are the **dominant modeling choice** in the robust-MDP and robust-RL literature. Most works that study robustness to transition uncertainty are formulated **directly in the MDP** setting rather than in POMDPs (e.g., [5][9]). Following this convention allows us to introduce a fundamental framework for efficient and robust policy evaluation that is aligned with prior work.
>
> We agree that extending robust behavior-policy search to partially observable environments is a promising direction. POMDPs introduce additional challenges such as belief-state uncertainty and latent transition ambiguity, and exploring such extensions would be a valuable next step. We appreciate the reviewer for highlighting this opportunity!
>
> 2. >For better readability and referencing (e.g., in Section 5.2), it would be beneficial to formally present key conditions using a numbered assumption environment.
>
> Thank you for this helpful suggestion. For readability and space considerations, we did not include a numbered assumption environment in the main text. That said, we agree that doing so would make the theoretical framework clearer, and we will incorporate numbered assumptions in the appendix of the revision.

---

> ### Author Response · Authors · 2025-11-22
>
> 3. >Could the authors provide intuition or a principled guideline for tuning $\eta$?
>
> Thank you for the thoughtful question. In our formulation, the coefficient $\eta$ controls the strength of the KL regularization in the inner-loop objective, and therefore the balance between robustness and fidelity to the simulator dynamics. In particular, smaller $\eta$ allows the adversarial transition to explore larger deviations, yielding stronger robustness, whereas larger $\eta$ keeps the adversarial transition closer to the original simulator transition $p_{\omega_0}$, preserving higher realism.
>
> This robustness–fidelity trade-off through the KL weight is standard in KL-regularized RL. For example, PPO explicitly adapts its KL penalty coefficient to target a desired divergence between successive policies [10], and [11] analyze how the KL coefficient influences the balance between robustness to approximation errors and convergence speed.
>
> In practice, the choice of $\eta$ depends on **how much trust** practitioners place in the simulator: the more reliable the simulator is believed to be, the larger $\eta$ can be set so that the adversary remains closer to the nominal transition model. Because this trade-off reflects task-specific modeling considerations, there is **no universal recommendation** for a single optimal value of $\eta$.
>
> Thank you again for these insightful questions and comments! We hope these clarifications resolve the questions raised, and we are happy to engage in any further discussion that may be helpful.
>
> **References**:
>
> [1] Moustafa Nouiehed, Meisam Razaviyayn.
> "Solving a Class of Non-Convex Min-Max Games Using Iterative First-Order Methods" (NeurIPS 2019)
>
> [2] Chi Jin, Praneeth Netrapalli, Michael I. Jordan.
> "What Is Local Optimality in Nonconvex–Nonconcave Minimax Optimization?" (ICML 2020)
>
> [3]Yifei Zheng, Samet Oymak, Yin Tat Lee.
> "Universal Gradient Descent–Ascent for Nonconvex–Nonconcave Minimax Problems" (NeurIPS 2023)
>
> [4] Alekh Agarwal, Sham Kakade, Jason Lee, Gaurav Mahajan.
> "On the Theory of Policy Gradient Methods: Optimality, Approximation, and Distribution Shift" (JMLR 2021)
>
> [5] Qiuhao Wang, Chin Pang Ho, Marek Petrik.
>      "Policy Gradient in Robust MDPs with Global Convergence Guarantee" (ICML 2023)
>
> [6] Josiah P. Hanna, Yash Chandak, Philip S. Thomas, Martha White, Peter Stone, Scott Niekum.
> "Data-Efficient Policy Evaluation Through Behavior Policy Search" (JMLR 2024)
>
> [7] Aviral Kumar, Aurick Zhou, George Tucker, Sergey Levine.
> "Conservative Q-Learning for Offline Reinforcement Learning" (NeurIPS 2020)
>
> [8] Aviral Kumar, Aurick Zhou, George Tucker, Sergey Levine.
> "Bootstrapping Error Accumulation Reduction (BEAR) for Offline Reinforcement Learning" (NeurIPS 2019)
>
> [9] Andrew Bennett, Nathan Kallus, Miruna Oprescu, Wen Sun, Kaiwen Wang.
>  "Efficient and Sharp Off-Policy Evaluation in Robust Markov Decision Processes" (NeurIPS 2024)
>
> [10] John Schulman, Filip Wolski, Prafulla Dhariwal, Alec Radford, Oleg Klimov.
> "Proximal Policy Optimization Algorithms" (PPO)
>
> [11] Nino Vieillard, Tadashi Kozuno, Bruno Scherrer, Olivier Pietquin, Rémi Munos, Matthieu Geist.
> "Leverage the Average: an Analysis of KL Regularization in Reinforcement Learning"  (NeurIPS 2020)
>
> [12] Zhongchang Sun, Sihong He, Fei Miao, Shaofeng Zou.
> "Policy Optimization for Robust Average Cost MDPs" (NeurIPS 2024)

---

> > ### Comment · Reviewer_eKd7 · 2025-11-25
> > **Reply to Authors' Rebuttal**
> >
> > Thank the authors for the detailed response. I agree that the minimax setup is a natural question in this area, and I see why adopting linear assumptions helps keep the analysis clean. At the same time, recent work has started to use measures like the Bellman eluder dimension [1], which can handle much richer function classes (including Neural Network) and could make the results feel more connected to real practical settings. It also tends to give a fairly clean theoretical story. I understand this may be outside the scope of the current paper, but it seems like a meaningful direction for future work.
> >
> > However, as the authors’ reply made the message of paper clearer, I am increasing my score to 6.
> >
> > [1] Jin, C., Liu, Q., & Miryoosefi, S. (2021). Bellman eluder dimension: New rich classes of rl problems, and sample-efficient algorithms. Advances in neural information processing systems, 34, 13406-13418.

---

> > > ### Author Response · Authors · 2025-11-25
> > >
> > > Thank you very much for the quick turnaround and for taking the time to reconsider the paper.
> > > We sincerely appreciate your thoughtful engagement during the rebuttal, especially your suggestions regarding related work and potential future directions such as Bellman eluder dimension and richer function classes. These comments are extremely valuable, and we will incorporate them to strengthen both the positioning and the broader impact of the work.
> > >
> > > Thank you again for your careful review and constructive feedback!

---

> ### Comment · Area_Chair_AXAp · 2025-11-25
> **From AC.**
>
> Hi reviewer eKd7,
>
> Do you find the author response convincing?
> I think it isn't very fair to ask an RL paper to resolve non-convexity of deep learning, would you agree?
>
> Thanks,
>
> area chair

---

> > ### Author Response · Authors · 2025-11-26
> >
> > Thank you for coordinating the discussion and for facilitating the exchange! We appreciate your time and effort in guiding the review process.

---

### Official Review · Reviewer_yyoQ · 2025-10-30

**Soundness:** 2
**Presentation:** 2
**Contribution:** 2
**Rating:** 2
**Confidence:** 4

**Summary:**

This paper addresses the challenge of high-variance policy evaluation in reinforcement learning (RL), especially when there is a mismatch between simulated and real-world transition dynamics. The authors propose a novel robust BPS framework formulated as a minimax optimization: an adversarial transition model is introduced to maximize the evaluation variance while the behavior policy is simultaneously optimized to minimize it. The core contribution is a double-loop gradient-based algorithm that alternates between finding worst-case (variance-maximizing) transition dynamics and updating the behavior policy to counteract those dynamics. The paper derives analytical expressions for the gradient of the evaluation variance with respect to transition parameters and provides theoretical guarantees of convergence for both the inner (adversarial transition) loop and the overall procedure. Empirically, experiments on benchmark tasks demonstrate that the proposed method achieves significantly lower evaluation variance under perturbed transitions compared to baseline approaches, indicating improved robustness to model mismatch.

**Strengths:**

Building upon prior work, the paper makes a notable breakthrough by integrating behavior policy search with adversarial robustness for the first time, specifically addressing the issue of environmental dynamics uncertainty that previous methods have largely overlooked. The innovation lies in introducing an adversarial transition model to simulate worst-case scenarios, thereby enabling the training of behavior policies that maintain low variance across diverse and uncertain dynamics.

**Weaknesses:**

Despite the aforementioned merits, the paper still exhibits several limitations that warrant the authors’ attention and improvement:
1. The proposed double-loop adversarial optimization framework enhances robustness but introduces additional computational complexity and overhead. The inner-loop adversarial training (which searches for the worst-case transition) and the outer-loop policy update are nested, requiring the simulation of a large number of trajectory samples per iteration. This leads to substantial computational costs, making the method potentially inefficient or impractical in large-scale or high-dimensional environments, thus constraining its real-world applicability.
2. The robustness of the proposed method is defined with respect to a pre-specified transition perturbation model—that is, the adversarial process searches within a parameter space Ω for the most adverse transition probability distribution. However, if the actual environmental variations fall outside the assumed uncertainty set (e.g., structural model biases not captured by the perturbation assumption), the robustness guarantees may degrade. In other words, the degree of robustness depends critically on whether the uncertainty set adequately encompasses real-world dynamics, a factor that must be carefully calibrated in practical deployments.
3. The authors assume that the importance sampling ratios (the probability ratio between the target and behavior policies in policy evaluation) are uniformly bounded, and that the transition probability function is continuously differentiable and twice differentiable with respect to its parameters. While these assumptions are mathematically convenient for deriving gradient formulations and convergence proofs, they may not always hold in practice. Relaxing these assumptions or analyzing the algorithm’s behavior when they are violated would improve the generality and theoretical robustness of the proposed approach.
4. Although the experimental results demonstrate the effectiveness of the proposed method, the evaluation remains somewhat limited in scope and depth. Specifically, experiments are conducted only on two medium-scale environments—Garnet MDPs and a simplified inventory management problem. These domains are relatively small and structurally simple, and thus may not fully capture the diversity and complexity of real-world applications.
5. While the authors compare their method against standard on-policy Monte Carlo evaluation and two existing behavior policy optimization approaches (e.g., BPG by Hanna et al. and the method by Zhong et al.), the empirical comparison could be further strengthened. In particular, although the related work section mentions the closed-form offline behavior policy solution by Liu and Zhang (2024) and other robust evaluation techniques (Katdare et al., 2023; Voloshin et al., 2021), these baselines are not included in the experimental comparison.

In summary, the paper presents an innovative and well-motivated solution addressing two key challenges in policy evaluation—variance reduction and environmental uncertainty. The contributions are substantial, and the results largely support the main claims. Nevertheless, the aforementioned issues—namely, the algorithmic complexity and assumptions, the limited experimental scope, and the incompleteness of baseline comparisons—should be further clarified and addressed.

**Questions:**

null

---

> ### Author Response · Authors · 2025-11-22
>
> We sincerely thank the reviewer for the thoughtful comments and constructive feedback. We are glad that the reviewer found our motivation compelling and recognized that the paper addresses key challenges in policy evaluation with substantial contributions. Below, we provide clarifications to fully address the remaining concerns.
>
>
> 1. >The proposed double-loop adversarial optimization framework enhances robustness but introduces additional computational complexity and overhead.
>
> Thank you for the question. Our setting reflects a common scenario where RL practitioners train both the target and behavior policies in a simulator, but must ultimately evaluate and deploy the policy in the real world. Because the simulator may not perfectly match reality, the final real-world evaluation can require a large number of costly real-world samples. Our double-loop adversarial optimization framework is specifically designed to **reduce real-world sample requirements** by proactively accounting for potential transition mismatch during simulator training.
>
> Naturally, there is a **trade-off**: incorporating robustness requires additional computation in the simulator stage compared with non-robust evaluation methods. However, this cost is incurred **only in the simulator**, where data are inexpensive and easily accessible.
>
> We also note that this double-loop adversarial structure is **standard** in robust RL. For example, Algorithm 1 in [1], Algorithm 1 in [2], and Algorithm 1 in [3] all adopt similar inner–outer optimization designs to incorporate robustness. These works face the same inherent trade-off: *investing extra (cheap) simulator computation to achieve better and more reliable real-world performance*. This principle—using abundant simulator data to anticipate adversarial real-world dynamics—is one of the core motivations behind robust RL.
>
> 2. >The robustness of the proposed method is defined with respect to a pre-specified transition perturbation model.
>
> We thank the reviewer for raising this important point. In robust RL, pre-specifying an uncertainty set for transitions is the standard modeling choice. This practice is used **throughout the robust MDP literature**—for example, ambiguity sets such as KL-balls, total-variation balls, or (s, a)-rectangular sets are adopted in [3][4][5][6]. These works all establish robustness relative to a user-specified uncertainty set, and this choice is not considered a limitation but a design component that allows practitioners to encode realistic transition uncertainties.
>
> In our setting, as is standard in the robust MDP literature, the robustness guarantee naturally depends on the uncertainty set $\Omega$, and choosing a richer class of perturbations simply strengthens coverage without changing the algorithmic framework. Thus, we believe our formulation is fully aligned with prior approaches and does not introduce additional modeling limitations beyond those commonly accepted in robust RL.
>
> 3. >The authors assume that the importance sampling ratios … are uniformly bounded, and that the transition probability function is continuously differentiable and twice differentiable with respect to its parameters. While these assumptions are mathematically convenient for deriving gradient formulations and convergence proofs, they may not always hold in practice.
>
> We appreciate the reviewer’s concern. Both assumptions are in fact *widely used* in the policy-gradient and off-policy evaluation literature. Specifically, the bounded importance-sampling ratio assumption is commonly adopted in behavior-policy search and importance-sampling analyses [7][8] to ensure well-defined variance objectives. And it can be trivially satisfied by ensuring the behavior policy $\pi_\theta$ is bounded away from zero.
>
> Likewise, assuming that the transition ​probability $p_\omega$ is continuously differentiable with bounded first- and second-order derivatives is a common smoothness assumption in theoretical RL (e.g., with similar assumptions used by [7][8][9]). Indeed, such assumptions are **not restrictive** in practice: when the transition model is parameterized with standard smooth neural network architectures (e.g., tanh, sigmoid, GELU) and a softmax output layer, the resulting function is automatically continuously differentiable with bounded first- and second-order derivatives, fully satisfying the conditions required for our analysis.

---

> ### Author Response · Authors · 2025-11-22
>
> 4. >These domains are relatively small and structurally simple.
>
> We appreciate the reviewer’s perspective. As the reviewer also noted, our work "presents an innovative and well-motivated solution addressing two key challenges in policy evaluation" . We introduce a new direction that unifies efficiency and robustness in policy evaluation through a novel minimax formulation, new analytical variance-gradient expressions for transitions, and the first global convergence result for behavior-policy search under adversarial dynamics. Our main contribution lies in the novel theoretical framework, and the experiments are intended as proof-of-concept validation demonstrating that the proposed method indeed yields improved robustness under transition perturbations. And as you suggested, our results "largely support the main claims", establishing a fundamental framework for future work to build upon.
>
> We would also like to note that the use of small to medium-scale environments is standard practice in papers establishing new theoretical frameworks in robust RL. For example, recent robust-RL papers such as [3][10] also use Garnet environments with smaller sizes than the one used by our paper, and some of them do not incorporate the classical stochastic-transition inventory management example. The comparison is summarized in Table 1 below.
>
> |               | Garnet | Inventory Management |
> |---------------|---------------|-------------|
> | **Ours**      |  G(30,15)      | Yes |
> | [3] (ICML 2023) | G(15,8)   |  Yes |
> | [10] (Neurips 2024)    | G(10,5)      | No |
>
> *Table 1:  Environments used in recent robust-RL works. Larger $G(\cdot,\cdot)$ settings reflect more challenging Garnet tasks, and our experiments operate at comparable or higher complexity.*
>
> 5. >In particular, although the related work section mentions the closed-form offline behavior policy solution by Liu and Zhang (2024) and other robust evaluation techniques (Katdare et al., 2023; Voloshin et al., 2021), these baselines are not included in the experimental comparison.
>
> We thank the reviewer for raising this point. We agree that it is important to clarify why certain methods were not included as baselines, and we provide detailed explanations below.
>
> **(1)** On the closed-form offline behavior-policy solution of Liu & Zhang (2024).
>
> Liu & Zhang (2024) operate in an **offline setting**, where a behavior policy is computed from a pre-logged dataset under fixed transitions. This differs fundamentally from our online behavior-policy search problem. Because their method requires specifying an offline dataset (choice of behavior policy, transition model, and coverage), it is unclear how to construct a **fair comparison** with online gradient-based approaches such as ours, BPG, or ROS (the two baselines we compare with). In addition, as noted in our paper, their method is tied to offline data under **fixed transitions** and cannot adapt to transition shifts, so it does not address the goal of designing a behavior policy that remains both variance-efficient and robust under adversarial transition perturbations.
>
> **(2)** On Katdare et al. (2023) and Voloshin et al. (2021).
>
> Both papers focus on improving the **accuracy** of policy evaluation under transition-model errors, whereas our work focuses on learning a robust behavior policy that minimizes evaluation **variance**. Specifically, Katdare et al. (2023) focus on off-environment policy evaluation using offline data and a simulator, aiming to correct evaluation bias rather than learning a data-collection policy. Voloshin et al. (2021) study minimax transition-model learning for robust model-based RL. Their goal is robustness of the model, not robustness of the behavior policy, and they do not address the variance-reduction or data-efficiency aspects central to behavior-policy search.
>
> In conclusion, these works address **different goals**—offline evaluation or robustness of model learning—rather than behavior-policy search for variance reduction. Since they are not designed to optimize behavior policies online, they are not directly comparable to our setting. We therefore include baselines (BPG, ROS) that target the same objective as our method.
>
> We hope these clarifications are helpful!  And we would gladly elaborate further if there are remaining questions or suggestions.

---

> ### Author Response · Authors · 2025-11-22
>
> **References**:
>
> [1] Cheng, Ching-An, Tengyang Xie, Nan Jiang, and Alekh Agarwal.
>
> "Adversarially Trained Actor Critic for Offline Reinforcement Learning" (ICML 2022)
>
> [2] Takumi Tanabe, Rei Sato, Kazuto Fukuchi, Jun Sakuma, Youhei Akimoto.
>
> "Max-Min Off-Policy Actor-Critic Method Focusing on Worst-Case Robustness to Model Misspecification" (NeurIPS 2022)
>
> [3] Qiuhao Wang, Chin Pang Ho, Marek Petrik.
>
> "Policy Gradient in Robust MDPs with Global Convergence Guarantee" (ICML 2023)
>
> [4] Yue Wang and Shaofeng Zou.
>
> "Online Robust Reinforcement Learning with Model Uncertainty" (NeurIPS 2021)
>
> [5] Andrew Bennett, Nathan Kallus, Miruna Oprescu, Wen Sun, Kaiwen Wang.
>
>  "Efficient and Sharp Off-Policy Evaluation in Robust Markov Decision Processes" (NeurIPS 2024)
>
> [6] Ho, Petrik, Wiesemann.
>
> "Partial Policy Iteration for L1-Robust MDPs" (JMLR 2021)
>
> [7] Josiah P. Hanna, Yash Chandak, Philip S. Thomas, Martha White, Peter Stone, Scott Niekum.
>
> "Data-Efficient Policy Evaluation Through Behavior Policy Search" (ICML 2017)
>
> [8] Josiah P. Hanna, Yash Chandak, Philip S. Thomas, Martha White, Peter Stone, Scott Niekum.
>
> "Data-Efficient Policy Evaluation Through Behavior Policy Search" (JMLR 2024)
>
> [9] Alekh Agarwal, Sham Kakade, Jason Lee, Gaurav Mahajan.
>
> "On the Theory of Policy Gradient Methods: Optimality, Approximation, and Distribution Shift" (JMLR 2021)
>
> [10] Zhongchang Sun, Sihong He, Fei Miao, Shaofeng Zou.
>
> "Policy Optimization for Robust Average Cost MDPs" (NeurIPS 2024)

---

> ### Comment · Area_Chair_AXAp · 2025-11-25
> **From AC,**
>
> Hi reviewer yyoQ!
>
> Do you find the author response convincing, particularly as regards the overhead of the method?
>
> Thanks,
>
> area chair

---

### Author Response · Authors · 2025-11-29
**Summary of Discussion Status**

Dear AC,

Thank you for taking the time to review our rebuttal. We wanted to provide a concise summary of the current review status to help you quickly understand the state of the discussion.

All three reviewers recognized that our work introduces a **novel and fundamental** framework for efficient and robust policy evaluation, and that the empirical results **consistently support** the theoretical developments. In our rebuttal, we addressed concerns regarding computational overhead, the strength of the theoretical assumptions, and the experimental scope relative to recent robust-RL literature.

After considering our responses, Reviewer eKd7 increased the score from 4 → **6**, and Reviewer qc4C maintained a **6** and explicitly **recommended acceptance**. Reviewer yyoQ has not yet replied, but our rebuttal directly addresses each of the concerns raised in their initial review.

We also greatly appreciated that the previous area chair expressed a **supportive attitude** toward our work and actively encouraged reviewers to consider the contribution of our approach.

Many thanks for your efforts in taking over the review process, and we are happy to provide any additional clarification that would be helpful.

Best regards,

The authors

---

### Meta-Review · Area_Chair_YGUa · 2025-12-19

**Summary:**

This paper introduces a novel double-loop adversarial optimization framework for robust behavior policy search under transition uncertainty, unifying variance reduction in off-policy evaluation with robust MDP principles. Reviewers acknowledge the work's theoretical originality and rigorous gradient derivations. However, significant flaws necessitate rejection.
The core issue is a substantial disconnect between theory and practice: the global convergence guarantee relies on a restrictive linear-softmax policy parameterization, while experiments use neural networks, invalidating the theoretical assurances. Furthermore, the proposed method is computationally expensive due to its nested adversarial optimization, and its empirical validation is limited to small-scale synthetic domains (Garnet MDP, inventory management), lacking demonstration on complex benchmarks. Critical baselines are omitted, and the robustness guarantees depend on strong assumptions (e.g., bounded importance ratios, well-specified uncertainty sets) that are not thoroughly discussed or relaxed. While the idea is promising, these unresolved theoretical, practical, and experimental limitations prevent the paper from making a convincing contribution in its current form.

**Reviewer Concerns:**

Reviewer yyoQ's concerns are still outstanding, and most of comments from reviewers eKd7 and qc4C were addressed.

**Reviewer Scores:**

Reviewer eKd7 will increase the score, while reviewers qc4C and yyoQ will maintain the scores.

---

### Decision · Program_Chairs · 2026-01-26

Reject